# Unlocking the function promiscuity of old yellow enzyme to catalyze asymmetric Morita-Baylis-Hillman reaction

Lei Wang[1,2,3], Yaoyun Wu ®[1,2,3], Jun Hu[1], Dejing Yin[2], Wanqing Wei ®[2,3], Jian Wen ®[1], Xiulai Chen ®[2,3], Cong Gao ®[2,3], Yiwen Zhou[1], Jia Liu[2,3], Guipeng Hu[1], Xiaomin Li[2,3], Jing Wu[1], Zhi Zhou ®[1], Liming Liu ®[2,3] & Wei Song ®[1] ✉

Exploring the promiscuity of native enzymes presents a promising strategy for expanding their synthetic applications, particularly for catalyzing challenging reactions in non-native contexts. In this study, we explore the promiscuous potential of old yellow enzymes (OYEs) to facilitate the Morita-Baylis-Hillman reaction (MBH reaction), leveraging substrate similarities between MBH reaction and reduction reaction. Using mass spectrometry and spectroscopic techniques, we confirm promiscuity of *Gk*OYE in both MBH and reduction reactions. By blocking H⁻ and H⁺ transfer pathways, we engineer *Gk*OYE.8, which loses its reduction ability but enhances its MBH activity. The structural basis of MBH reaction catalyzed by *Gk*OYE.8 is obtained through mutation studies and kinetic simulations. Furthermore, enantiocomplementary mutants *Gk*OYE.11 and *Gk*OYE.13 are obtained by directed evolution, exhibiting the ability to accept various aromatic aldehydes and alkenes as substrates. This study demonstrates the potential of leveraging substrate similarities to unlock enzyme functionalities, enabling the catalysis of new-to-nature reactions.

C-C bond formation is a crucial process in organic synthesis, playing a key role in establishing the carbon backbone of organic molecules[1]. Nevertheless, the limited functionality of native enzymes impedes their widespread application in C-C bond formation. Hence, the expansion of enzymatic catalytic functions holds paramount significance in accelerating the progress of biomanufacturing, representing a fundamental objective in both academia and industry[2]. Presently, various methods, such as protein engineering[3–5], artificial enzymes[6–8], and computer-aided de novo design[8,9], have been developed to expend the catalytic functions of enzymes. Fundamentally, these approaches underscore the leveraging of enzymes' broad promiscuity to achieve innovative functionalities.

Utilizing enzyme promiscuity (including condition promiscuity, substrate promiscuity, and catalytic promiscuity) to develop unnatural bond-forming functions of native enzymes has been widely studied.

Firstly, new C-C bond-forming functions can be induced by altering catalytic reaction conditions, such as visible light. For example, to achieve C($sp^2$)-C($sp^3$) bond formation, Xiaoqiang Huang group employed visible light to excite flavin-dependent ene reductase (naturally catalyzing the double-electron reduction of alkenes) for conducting redox-neutral asymmetric radical hydroarylation reactions, with 81% yield and (*R*)-preferred selectivity (97.5:2.5 e.r.), 60% yield with (*S*)-preferred selectivity (90:10 e.r.) realized by different ene-reductases with model substrates[2]. Noteworthy examples also include utilizing photoexcited ene reductase for ene-allylation reactions[10] and radical hydrogenation reactions[11]. Secondly, enzyme substrate promiscuity refers to their broad substrate specificity, a notable example involves using tryptophan synthase TrpB to produce various non-natural amino acids (ncAAs)[12,13]. To catalyze non-enantioselective C-C, C-N, and C-S bond formation reactions, the TrpB from *Pyrococcus*

[1]School of Life Sciences and Health Engineering, Jiangnan University, Wuxi 214122, China. [2]School of Biotechnology, Jiangnan University, Wuxi 214122, China. [3]Key Laboratory of Industrial Biotechnology of Ministry of Education, Jiangnan University, Wuxi 214122, China. ✉e-mail: weisong@jiangnan.edu.cn

*furiosus* was engineered through directed evolution, the mutant *Pf*TrpB[2B9], for instance, could one-pot synthesize (2*S*,3*S*)-β-methyl-tryptophan (β-MeTrp) and various indole analogs and thiophenes, which enabled >99% conversion of indole to β-MeTrp and up to 8200 total turnovers to the desired product with >99% ee and de[14]. Thirdly, enzyme catalytic promiscuity refers to the ability of an active site to catalyze different chemical reactions. A representative example is expanding the native C-O bond formation function of P450s to catalyze carbene transfer reactions. To catalyze cyclopropanation reactions on unactivated olefinic substrates and olefinic substrates with heteroatom substitution, cytochrome P411 was engineered, and its variants were obtained with high diastereoselectivity (97:3 dr) and enantioselectivity (97% ee) to synthesize cyclopropane compounds[15]. Other C-C bond formation examples based on enzyme catalytic promiscuity include lipases catalyzing Aldol and Michael addition reactions[16,17], and ThDP-dependent enzymes catalyzing decarboxylation C-C bond-forming reactions[18]. The above cases strongly demonstrate that expanding enzyme promiscuity is a valuable approach to endow native enzymes with new catalytic functions, fostering advancements in the field of C-C bond formation.

The Morita-Baylis-Hillman (MBH) reaction, a typical C-C bond-bonding reactions, refers to an atom-economic transformation wherein an activated alkene (e.g., *α*, *β*-unsaturated carbonyl compounds) reacts with a carbon electrophile (e.g., aldehyde) to form adducts. The resulting MBH adducts have demonstrated diverse biological activities, including anticancer, antidiabetic, anti-inflammatory, antiviral, antibacterial and others[19]. Therefore, various chemical catalysts are employed for MBH reaction (Fig. 1a), including nitrogen-based, phosphorus-based, chalcogen-based Lewis base catalysts, and Lewis acid catalysts like TiCl₄ and Et₂AlI, as well as multi-catalyst systems[20]. However, the chirality of available catalysts is highly dependent on the structure of substrates, limiting the widespread application of MBH reaction. In nature, no native enzymes have been discovered to efficiently catalyze MBH reaction. Only a few enzymes, such as serum albumin and lipases, exhibit low levels of promiscuous activity (2%-35% conversion) towards MBH reaction[21]. The maximum enantiomeric excess achieved for different bovine serum albumins (BSA) was 19%. When lipases catalyze the MBH reaction, in addition to the issue of low yield (15%), a significant amount (80%) of aldol by-products is generated[22]. To address this, there developed an artificial MBHase, BH32.14, through a combination of computational design and directed evolution[23]. As shown in Fig. 1b, BH32.14 exhibited high catalytic efficiency (94% conversion) and excellent stereoselectivity (93% ee). Unfortunately, only a single configuration of MBH adducts can be stimulated at present. Furthermore, it is challenging that the applicability of de novo protein design techniques and differences between the design models and real protein structures, extensive mutagenesis work is required in later stages.

In this study, we explore the development of a promiscuous function for old yellow enzymes (OYEs) in catalyzing MBH reaction, based on the substrate similarity between the MBH reaction and its native reduction reaction (Fig. 1c). Firstly, it is determined that the old yellow enzyme *Gk*OYE possesses a promiscuous function in catalyzing both reduction reactions and the MBH reaction. Subsequently, the

**a. Chemical approaches of MBH reaction.**

X=O, NR² 
R₁=alkyl, aryl or heteroaryl

EWG=CO₂R³, NO₂, CHO, C(O)R³, etc

MBH adducts

**b. MBHase created by computational design and directed evolution.**

R=aryl or heteroaryl

BH32.14 
PBS pH 7.4 30 ℃

**c. This study.**

Native reduction function :

C=C reduction

C-C carboligation

MBH function :

*Gk*OYE

*Gk*OYE.11 
(*R*)-MBH adducts

*Gk*OYE.13 
(*S*)-MBH adducts

**Fig. 1 | The chemical and biocatalytic approaches of MBH reaction and strategy of this study. a** The chemical approaches of MBH reaction. **b** Biocatalytic approaches of MBH reaction designed by computational design and directed evolution. **c** Strategy of utilizing the promiscuity of OYEs to achieve C-C bond-forming reactions in this study.

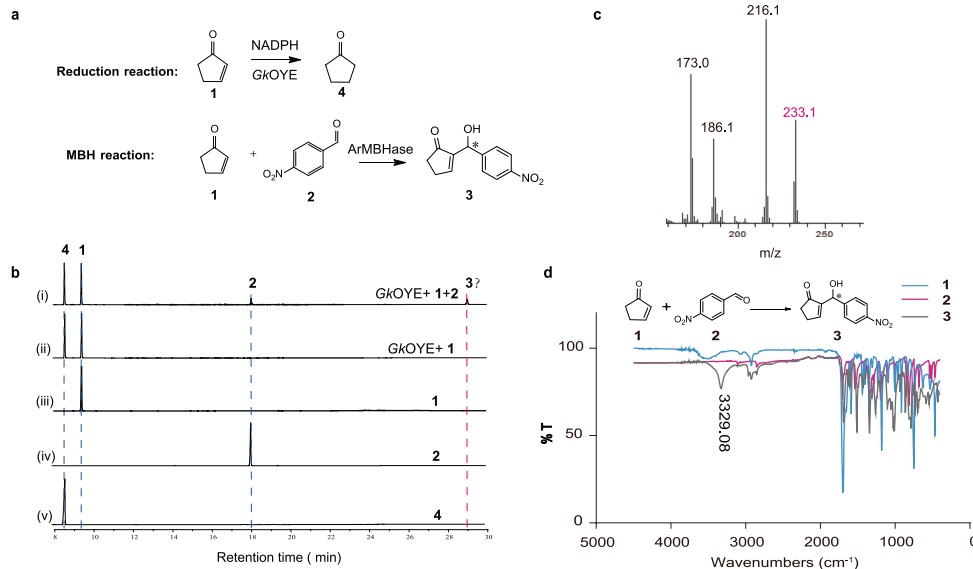

**Fig. 2 | Detection and identification the new substances in the reaction systems.** **a** The equations involved in this study. **b** GC analysis detected the formation of a new substance, labeled as compound **3**. The reactions were carried out using 5 mM **1**, 1 mM **2**, 1 mM NADPH, and 100 μM GkOYE in PBS (pH 7.4) with 3% methanol (MeOH) as a co-solvent. **c** The results of GC-MS. The molecular weight corresponding to characteristic absorption peaks of the product is determined to be at m/z = 233.1. **d** The results of infrared spectroscopy. The data exhibited a characteristic absorption peak of hydroxyl groups at 3329.08 cm$^{-1}$.

native reduction function of GkOYE is eliminated by blocking the H$^{-}$ and H$^{+}$ transfer pathway, while the function catalyzing MBH reaction is enhanced by 141.4% higher than GkOYE. The structural basis for the occurrence of the MBH reaction is elucidated through mutagenesis studies and kinetic simulations. Finally, further protein engineering efforts are undertaken to improve the catalytic efficiency and stereoselectivity of the reaction, leading to the synthesis of both (R)-MBH adducts and (S)-MBH adducts.

## Results

### Identifying function promiscuity of GkOYE to catalyze MBH reaction

Since no native MBHases have been found yet, this study aims to expand the function promiscuity of other native enzymes to facilitate the MBH reaction. In this study, commonly used 2-cyclopenten-1-one (**1**) and 4-nitrobenzaldehyde (**2**) were chosen as model substrates for the MBH reaction (Fig. 2a). According to the substrate similarity, substrate **1** (an α, β-unsaturated ketone) is typically used as a native substrate for the asymmetric C = C bond reduction catalyzed by old yellow enzymes (OYEs)[24,25]. Therefore, we hypothesized that a function for catalyzing MBH reaction can be developed based on the alkene reduction function (native function) of OYEs. Surprisingly, in addition to reducing the C = C bond of **1** to generate product **4** (Fig. 2a), there detected a new compound (Fig. 2b) when substrate **2** was added to GkOYE (OYE from thermophilic bacterium *Geobacillus kaustophilus*; PDB: 3gr7) catalyzed reaction system. To identify the structure of this compound, it was purified by preparative high performance liquid chromatography and analyzed by GC-MS, IR, and NMR. As shown in Fig. 2c, GC-MS results indicated that the relative molecular mass of this compound was 233.1, matching the theoretical molecular weight of MBH product **3** (233.2). While IR data exhibited a characteristic absorption peak of hydroxyl groups at 3329.08 cm$^{-1}$ (Fig. 2d), which is also consistent with the structure of product **3**. Finally, the $^{1}$H and $^{13}$C NMR spectra of this newly formed compound were confirmed as the characteristics of target product **3** (Supplementary Fig. 1).

To identify which component(s) in the above reaction system really catalyzed the reaction, the influence of the components (including solution buffer, NADPH, FMN, and GkOYE protein) on the MBH reaction was investigated. Substrates **1** and **2** incubated with

solution buffer was set as control (Table 1, entry 1), which exhibited only a weak background reaction, resulting in 8.4 μM of **3**. Subsequently, NADPH, FMN, or GkOYE protein was added separately into buffer, and the titer of **3** was detected. As shown in Table 1, adding GkOYE protein generated 44.3 μM of **3** (Table 1, entry 4), significantly higher than the experimental groups with NADPH (8.5 μM; Table 1, entry 2) or FMN (5.7 μM; Table 1, entry 3). This result indicated that it is indeed the GkOYE protein that catalyzes the MBH reaction in the reaction system. It's worth noting that, adding NADPH had no significant effect on the reaction, while the addition of FMN reduced titer of **3** by 32.1% compared to control, suggesting that FMN may have an inhibitory effect on MBH reaction. To confirm this conclusion, NADPH and FMN were added separately to the system containing GkOYE protein. The results showed that NADPH (Table 1, entry 5) had no significant effect, while FMN (Table 1, entry 6) indeed had an inhibitory effect, reducing the titer of **3** by 28.7% (from 44.3 μM to 31.6 μM). These results demonstrate that the GkOYE exhibits promiscuity, which could catalyze both native reduction reactions and the MBH reaction, which was not reported previously. To investigate whether other OYEs

**Table 1 | The MBH reactions of 1 and 2 catalyzed by different catalysts**

| Entry | Catalyst | Catalyst loading (mM) | Titer (μM) |
|---|---|---|---|
| 1 | None | None | 8.4 |
| 2 | NADPH | 1 | 8.5 |
| 3 | FMN | 1 | 5.7 |
| 4 | GkOYE | 0.1 | 44.3 |
| 5 | GkOYE + NADPH | a | 45.8 |
| 6 | GkOYE + FMN | b | 31.6 |
| 7 | NemA | 0.1 | 35.5 |
| 8 | XenA | 0.1 | 35.7 |
| 9 | GluER | 0.1 | 40.4 |
| 10 | MR | 0.1 | 43.9 |

Entries 1–10 were carried out using **1** (5 mM), **2** (1 mM) in PBS (pH 7.4) with 3% methyl alcohol as a co-solvent. Conversion to product was determined by HPLC analysis. $^{a}$ Catalyst loading of GkOYE was 0.1 mM with 1 mM NADPH. $^{b}$ Catalyst loading of GkOYE was 0.1 mM with 1 mM FMN.

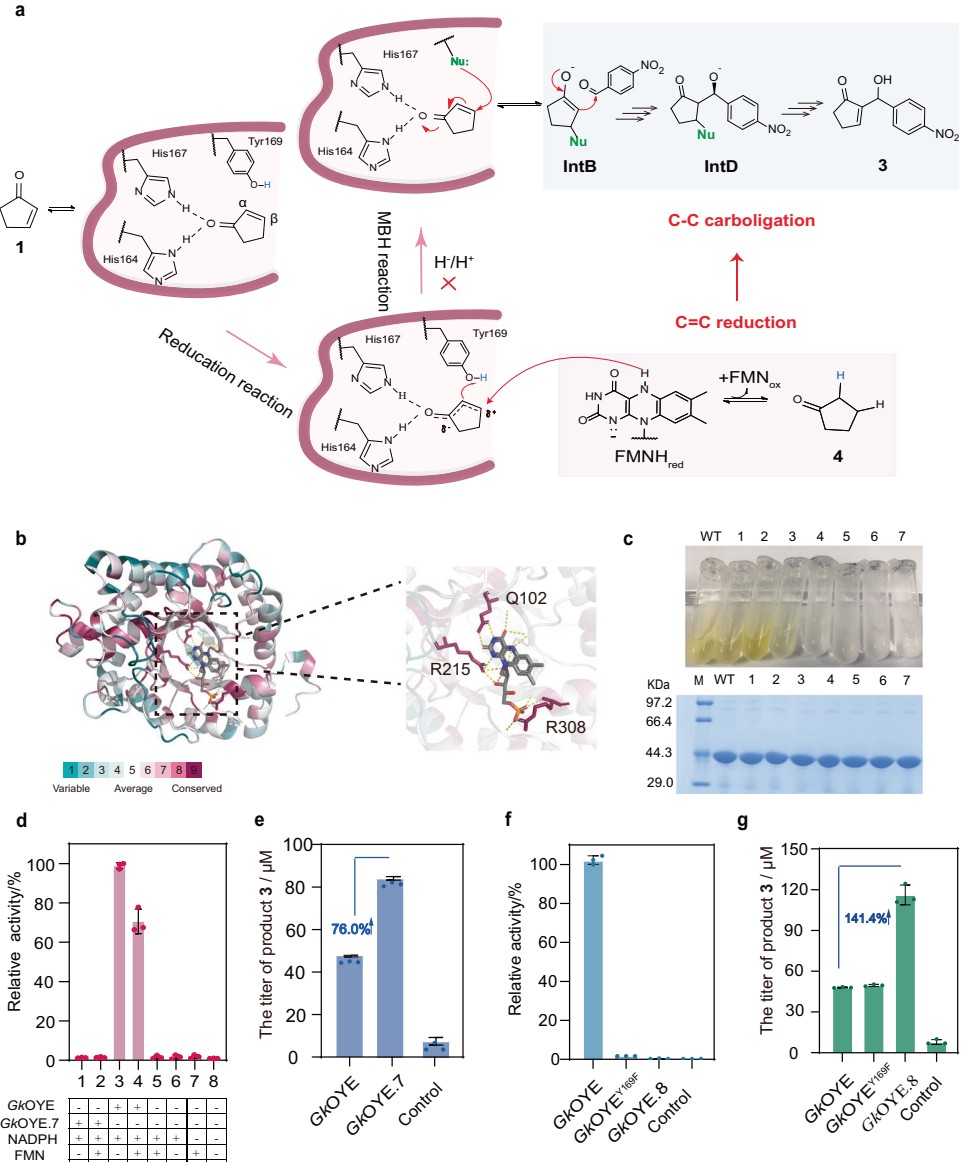

**Fig. 3 | Enhancing MBH function of *Gk*OYE from the perspective of catalytic mechanism. a** The mechanism of natural reduction reaction catalyzed by OYEs and the putative mechanism of MBH reaction in this study. The mechanism of MBH reaction was inferred based on the reported mechanisms of enzyme catalysis[23] and chemical small molecule catalysis[56]. Nu: The nucleophilic catalytic residue in *Gk*OYE protein. **IntD**: The covalent intermediates are formed by the nucleophilic catalytic residue Nu with the βC of the double bond in substrate **1** and the aldehyde group C in substrate **2**. **b** Conservation analysis of the residues interacting with FMN in the *Gk*OYE protein. **c** Color and SDS-PAGE validation for wild-type and mutant protein. Note: 1-7 correspond to different variants of the protein. 1: *Gk*OYE.1、 2: *Gk*OYE.2、 3: *Gk*OYE.3、 4: *Gk*OYE.4、 5: *Gk*OYE.5、 6: *Gk*OYE.6、 7: *Gk*OYE.7. Three times each experiment was repeated independently with similar results. **d** Comparison of

reduction activity between *Gk*OYE and variant *Gk*OYE.7. Columns 1 and 2 represent the reduction activity of *Gk*OYE.7 with and without FMN. Columns 3 and 4 represent the reduction activity of *Gk*OYE with and without FMN. Columns 5-8 represent blank control groups. + indicates presence and - indicates absence. The added components are 5 mM **1**, 1 mM NADPH, 1 mM FMN, and 100 μM purified protein in PBS (pH 7.4) with 1% methanol (MeOH) as a co-solvent. **e** Comparison of catalytic activity of MBH reaction between *Gk*OYE and variant *Gk*OYE.7. The reactions were carried out using 5 mM **1**, 1 mM **2**, and 100 μM purified protein in PBS (pH 7.4) with 3% methanol (MeOH) as a co-solvent. **f** Validation of reduction activity for different mutants. **g** Comparison of catalytic activity for MBH reaction between different mutants. *n* = 3 independent biological experiments. Data are presented as mean values ± SD. Source data are provided as a Source Data file.

also have similar promiscuity, four widely used OYEs[26], including NemA from *E. coli* (Table 1, entry 7), XenA from *P. Putida* (Table 1, entry 8), GluER from *G. Oxydans* (Table 1, entry 9), and MR from *P. Putida* (Table 1, entry 10), were selected and tested (Supplementary Fig. 2). All these four OYEs were able to catalyze the MBH reaction, and the titer of **3** (35.5, 35.7, 40.4, and 43.9 μM, respectively) was significantly higher than the control (8.4 μM). In summary, the MBH function is universal in OYE family. As *Gk*OYE exhibited a higher titer of **3** than other four OYEs, it has been selected as the preferred enzyme for subsequent experiments.

## Blocking hydrogen transfer pathways to enhance MBH function of *Gk*OYE

In above experiment (Table 1, entry 5), although 45.8 μM of **3** was produced, there also generated 6.3 folds of reduction product **4** simultaneously (see Supplementary Fig. 3), indicating *Gk*OYE prefers the native C = C reduction reaction. To eliminate the native reduction function and enhance the C-C bond-forming function for MBH reaction, two hydrogen transfer pathways were disrupted based on the mechanism of OYE catalyzed C = C reduction. As illustrated in Fig. 3a, the two hydrogen transfer pathways for reduction reaction are as

follows: (1) The N5 position of the reduced FMN molecule transfers an H$^-$ to the βC of the substrate, and (2) the hydroxyl group of catalytic residue Y169 transfers an H$^+$ to the αC of the substrate[27]. Both H$^-$ and H$^+$ transfer pathways must act in concert for the reduction reaction to proceed. Therefore, the C = C reduction function can be eliminated by disrupting the H$^-$ and H$^+$ transfer pathways. Hence, the following approaches were proposed: (1) removing the cofactor FMN to disrupt H$^-$ transfer, (2) mutating Y169 to disrupt H$^+$ transfer, and (3) disrupting H$^-$ and H$^+$ transfer pathways simultaneously.

To remove the cofactor FMN and thus block H$^-$ transfer, a site-specific mutation was employed to break the interaction between GkOYE and FMN. It is reported that FMN combines with OYEs in a non-covalent manner, and 17 residues (Supplementary Fig. 4) have been identified to interact with FMN in related OYEs (e.g. BsYqjM)[28]. Further evolutionary conservation analysis revealed that, among these 17 residues, Q102, R215, and R308 were relatively conserved (Fig. 3b) and had more interactions with FMN, making them key residues for binding FMN. These three key residues were subjected to alanine scanning and combinatorial mutagenesis to disrupt the interaction between GkOYE and FMN, resulting in seven mutants, including GkOYE.1 (GkOYE$^{Q102A}$), GkOYE.2 (GkOYE$^{R215A}$), GkOYE.3 (GkOYE$^{R308A}$), GkOYE.4 (GkOYE$^{Q102A/R215A}$), GkOYE.5 (GkOYE$^{Q102A/R308A}$), GkOYE.6 (GkOYE$^{R215A/R308A}$), and GkOYE.7 (GkOYE$^{Q102A/R215A/R308A}$). As shown in Fig. 3c, the results indicated that the purified protein solutions of GkOYE, GkOYE.1, GkOYE.2, and GkOYE.3 were yellow, indicating FMN was not removed. While the protein solutions of GkOYE.4, GkOYE.5, GkOYE.6, and GkOYE.7 were colorless and transparent, and detected no FMN peaks using LC-MS (Supplementary Fig. 5), indicating FMN was successful removed from them. Additionally, GkOYE.7 exhibited the largest pocket volume among these four mutants (Supplementary Table 1), which facilitates the binding of substrate 2 in the pocket. Therefore, GkOYE.7 was chosen for subsequent functional validation. The result showed that the native reduction function of GkOYE.7 was lost (Fig. 3d), and GkOYE.7 was unable to catalyze the reduction reaction regardless of whether FMN was added to the reaction system. While, the titer of MBH adduct 3 (84.5 μM) generated by GkOYE.7 was 76.0% higher than that of wild-type GkOYE (Fig. 3e), indicating that blocking the transfer of H$^-$ successfully enhanced MBH reaction.

To block the transfer of H$^+$ from Y169 to substrate 1, Y169 was mutated to Phe, resulting in mutant GkOYE$^{Y169F}$. Subsequently, the reduction and MBH functions of GkOYE$^{Y169F}$ were validated. As shown in Fig. 3f, the reduction activity of GkOYE$^{Y169F}$ decreased by 98.3% compared to wild-type GkOYE, demonstrating that GkOYE$^{Y169F}$ basically lost its native reduction function. However, GkOYE$^{Y169F}$ only produced 50.3 μM of MBH adduct 3, slightly higher (3.3%) than wild-type and 41.4% lower than GkOYE.7 (Fig. 3g). Based on this, the mutation Y169F was introduced into GkOYE.7, simultaneously disrupting H$^+$ and H$^-$ transfer, resulting in mutant GkOYE.8 (GkOYE$^{Q102A/Y169F/R215A/R308A}$). Surprisingly, GkOYE.8 could produce 116.0 μM of MBH adduct 3, which was 141.4% higher than GkOYE and 37.2% higher than GkOYE.7. In summary, by blocking H$^+$/H$^-$ transfer, the functional selectivity of GkOYE was reversed, which disrupted its native C = C reduction function and enhanced MBH reaction.

**Revealing structural bases of GkOYE.8 catalyzed MBH reaction**
It is of important to reveal the structural bases of GkOYE.8 catalyzed MBH reaction. Therefore, we first tried to obtain the GkOYE.8-substrate complex structure through X-ray crystallographic analysis, to clarify the substrate binding site for a better understanding of catalytic residues. However, only apo-GkOYE.8 structure was obtained with resolution of 3.11 Å (Supplementary Fig. 6, Supplementary Table 2). As shown in Fig. 4a, apo-GkOYE.8 forms a dimeric structure, with each monomer adopting the classic TIM barrel structure consisting of 8 α-helices and 8 β-sheets. A comparison with GkOYE-FMN complex structure (PDB ID: 3gr7) revealed an RMSD value of 0.241, and only

His164 exhibited a slight deviation within the pocket (Fig. 4b)[29]. This indicates that the structure of apo-GkOYE.8 protein does not undergo significant changes in the absence of FMN binding. Notably, we found the enantiomeric ratio of GkOYE.7 and GkOYE.8 was different, of which GkOYE.7 yielded (R)-3 with 53:47 e.r., while GkOYE.8 achieved an e.r. value of 66:34 (Fig. 4c). It can be concluded that F169 significantly affects the configuration of adduct 3, suggesting MBH reaction center is located near F169 within the catalytic pocket, rather than in other places of protein. Then, molecular docking was performed to obtain the GkOYE.8-substrate complex structure. Here, the structure of adduct 3 was similar to that of IntD (Fig. 4d), which was used as the docking ligand. Based on F169, we assumed that the carbonyl oxygen of 3 is still anchored by hydrogen bonding interactions with H164 and H167[30,31], and obtained the complex structure of GkOYE.8-3 (Fig. 4d).

To identify the key residues involved in the MBH reaction, alanine scanning was performed on the 17 residues within 4 Å radius of 3 (Fig. 4d, excluding A60 and A252). As shown in Fig. 4e, the mutant H167A exhibited a 38.5% increased yield compared to GkOYE.8, the other mutants had varying degrees of decreased yields. Among them, mutants with a yield reduction of over 50% included P24A, C26A, E59A, I69A, K109A, E162A, H164A, and D247A. The most significant yield reduction was observed in mutants C26A, E59A, and H164A, which showed a yield decrease of over 70%, suggesting they are key residues that make up the active center for the MBH reaction. Circular dichroism scans (Supplementary Fig. 7) revealed that the secondary structure of mutants C26A, E59A, and H164A did not change significantly compared to GkOYE.8. This suggests that the reduced yield is not a result of alterations in protein structure but rather stems from the direct impact of these residues on the MBH reaction. To further investigate the roles of C26, E59, and H164 in MBH reaction, saturation mutagenesis was conducted. Significantly, when C26 was mutated to other 19 residues, the yield of adduct 3 decreased by more than 60% (Fig. 4f), suggesting C26 is a key residue for catalyzing MBH reaction. Therefore, C26 may be a nucleophilic catalytic residue involved in the MBH reaction[32] that attacks βC of substrate 1 to form a covalent IntB (Fig. 3a). Sequence alignment analysis was conducted on the five old yellow enzymes listed in Table 1. The results indicate that residue C26 is conserved in NemA, XenA, and GkOYE. In GluER and MR, residue 26 is Thr, which can also act as a nucleophile attacking the βC of the double bond of substrate 1 (Supplementary Fig. 8). All E59 mutants showed significantly reduced protein solubility and were mostly found in inclusion bodies, indicating that E59 is crucial for maintaining the protein tertiary structure (Supplementary Fig. 9). As to H164, mutated H to W, K, A, and G led to 50%-70% decreases in 3 yield, and mutated to other residues retains production of 3 over 60% (Fig. 4g). These results suggest that His164 is not directly involved in the MBH reaction but may be related to support a suitable catalytic conformation.

To determine whether C26 is the nucleophilic residue, protein mass spectrometry was recruited to detect the accumulation of intermediate B. As shown in Fig. 5a, incubating GkOYE.8 with substrate 1 resulted in a 2-cyclopentenone moiety (84.5 Da) labeled GkOYE.8, indicating the presence of intermediate B. In contrast, no accumulation of intermediate B was observed when using GkOYE.8$^{C26A}$ to replace GkOYE.8 (Fig. 5b), providing strong evidence that C26 indeed acts as a nucleophilic residue to catalyze MBH reaction. Furthermore, quantum mechanics (QM) calculations employing density functional theory (DFT) also suggest that residues C26 and E59 may be key residues in the reaction process. Given the composition of the reaction system, we therefore compared the energies of the two proton transfer modes by DFT calculations (Supplementary Fig. 10) and the calculations suggested that phosphate anion may facilitate proton transfer in the third and fourth chemical steps. As shown in Fig. 5c, in the background reaction without enzyme, the transition state [**TS0**] and [**TS1**] have the higher energy barriers (12.4 kcal/mol and 14.5 kcal/mol, respectively). However, in the theozyme model with the addition of the catalytic

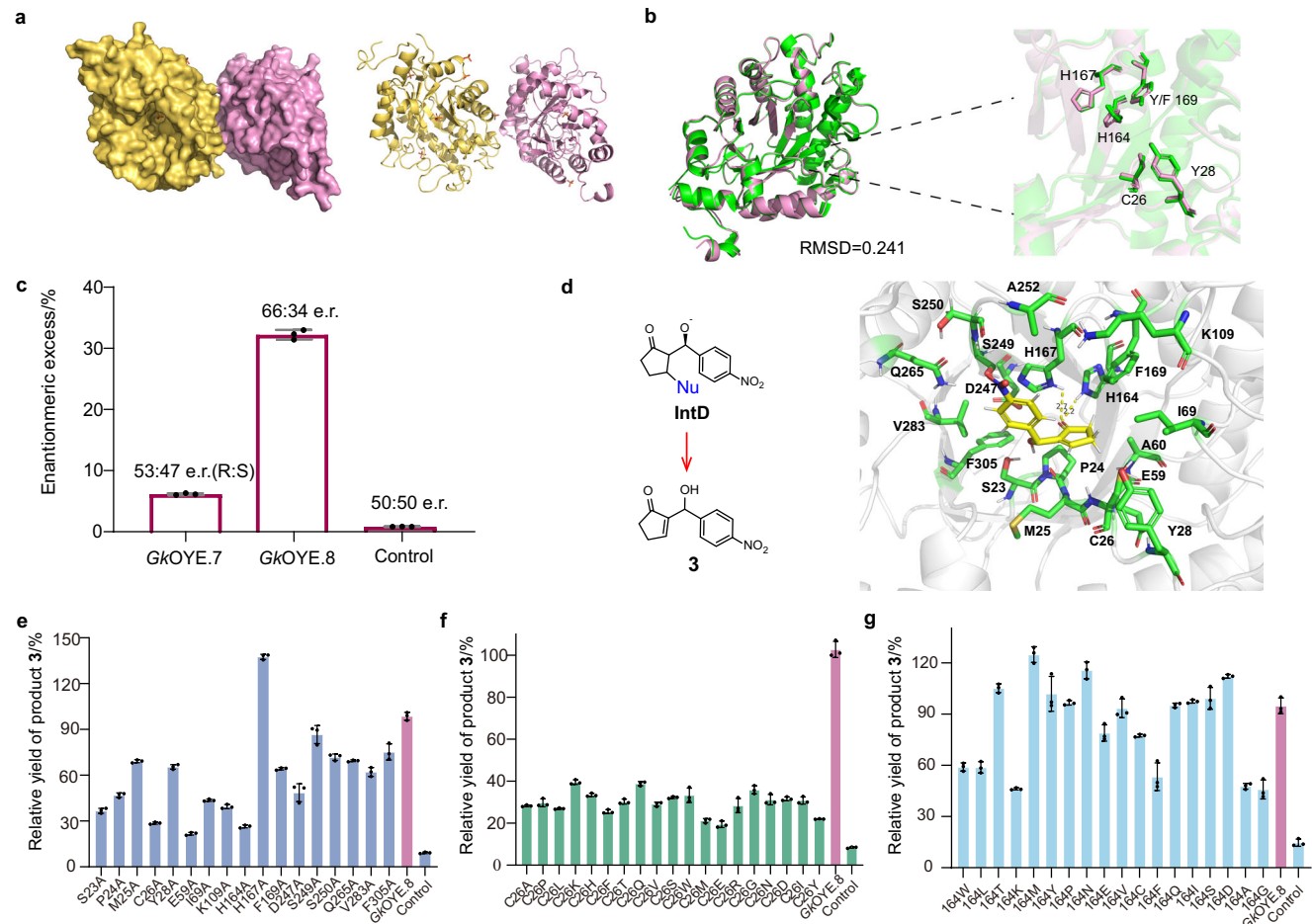

**Fig. 4 | Investigating the structural basis of MBH reaction catalyzed by** ***Gk*OYE.8. a** The dimer architecture of *apo-Gk*OYE.8 shown as surface pattern and cartoon pattern. **b** Structure alignment of *Gk*OYE (PDB ID: 3gr7) and *apo-Gk*OYE.8 (PDB ID: 8X0J). *Gk*OYE shown as green, *apo-Gk*OYE.8 shown as pink. **c** Determination of stereoselectivities of *Gk*OYE.7 and *Gk*OYE.8. **d** The architecture of the complex form of *Gk*OYE.8-**3** and the key residues within a 4 Å radius. **e** The results of the alanine scan. **f** The results of C26 sites saturation mutation. **g** The results of H164 sites saturation mutation. $n = 3$ independent biological experiments. Data are presented as mean values ± SD. Source data are provided as a Source Data file.

residues C26 and E59 (Fig. 5d), the higher energy barrier steps [**TS0'**] and [**TS1'**] remain unchanged, but their energy barriers drop to 11.0 kcal/mol and 12.1 kcal/mol, respectively, so it can be concluded that residues C26 and E59 improve the catalytic efficiency of the reaction by reducing the energy barriers of these two higher energy barrier steps. In addition, the protonation state (p$Ka$ value) of C26 thiol is a crucial determinant of cysteine nucleophilicity[33]. Typically, the protonation state of thiol is affected by the local micro-environment of surrounding residues and the pH of the reaction system. To explore whether other residues, typically acidic residues, assist in removing H+ from the C26 thiol during MBH reaction, a 100 ns restrained MD simulation (constrained the distance between sulfur atom of C26 and C6 of **3** with a harmonic potential having an equilibrium length of 3.65 Å) was performed on *Gk*OYE.8-**3** complex. Within a 4 Å radius of C26, five residues were detected (highlighted in green), with E59 being the sole acidic residue (Fig. 5e). During the MD simulation, a water molecule was observed between C26 and E59, creating a water-mediated hydrogen bond network that facilitated the removal of H+ from the C26 thiol group by E59. Among 5000 conformations output by MD, approximately 6.1% displayed this water-mediated H+ removal configuration. This implies that the reaction center may comprise both C26 and E59, with C26 acting as the nucleophilic agent for direct covalent catalysis with substrate **1**, while E59 is responsible for enhancing the nucleophilicity of C26 by facilitating H+ removal through a water-mediated process. On the other hand, PROPKA

calculations indicated a p$Ka$ of 12.25 for C26, far from the environmental pH (7.4), suggesting a weak nucleophilicity since cysteine is predominantly in the -SH form in this reaction pH. Raising reaction pH from 6.0 to 10.0 enhanced the nucleophilic nature of cysteine, leading to increased yield of adduct **3** (Fig. 5f), and the highest yield (32.3%) was obtained at pH 10.0. Raising reaction pH to 11.0 induced the chemical MBH reaction directly in strongly alkaline environment, resulting in a tenfold increase in background reaction within the control group. In this case, stereoselective synthesis of (*R*)-**3** cannot be achieved. Furthermore, we investigated the effects of carbonate buffer and glycine-NaOH buffer on the MBH reaction catalyzed by *Gk*OYE.8. The results showed that while both buffers yielded higher MBH product **3** compared to PBS buffer, they also led to significantly higher background reaction rates (yielding 51.4% and 46.0%, respectively). This high background reaction resulted in a substantial reduction in the enantiomeric ratio of (*R*)-**3** to 50:50 (Supplementary Fig. 12). Therefore, we concluded that PBS buffer at pH 10.0 is more suitable for *Gk*OYE.8 catalyzed MBH reactions. In this setting, the presence of hydroxide ions (OH-) in the environment aids in extracting H+ from the C26 thiol group, thereby enhancing the MBH reaction.

**Engineering *Gk*OYE to improve catalytic performance for MBH reaction**

Due to the low yield (32.3%) and enantiomeric ratio ((*R*)-**3**:(*S*)-**3** = 66:34 e.r.) of *Gk*OYE.8, directed evolution was adopted to enhance the

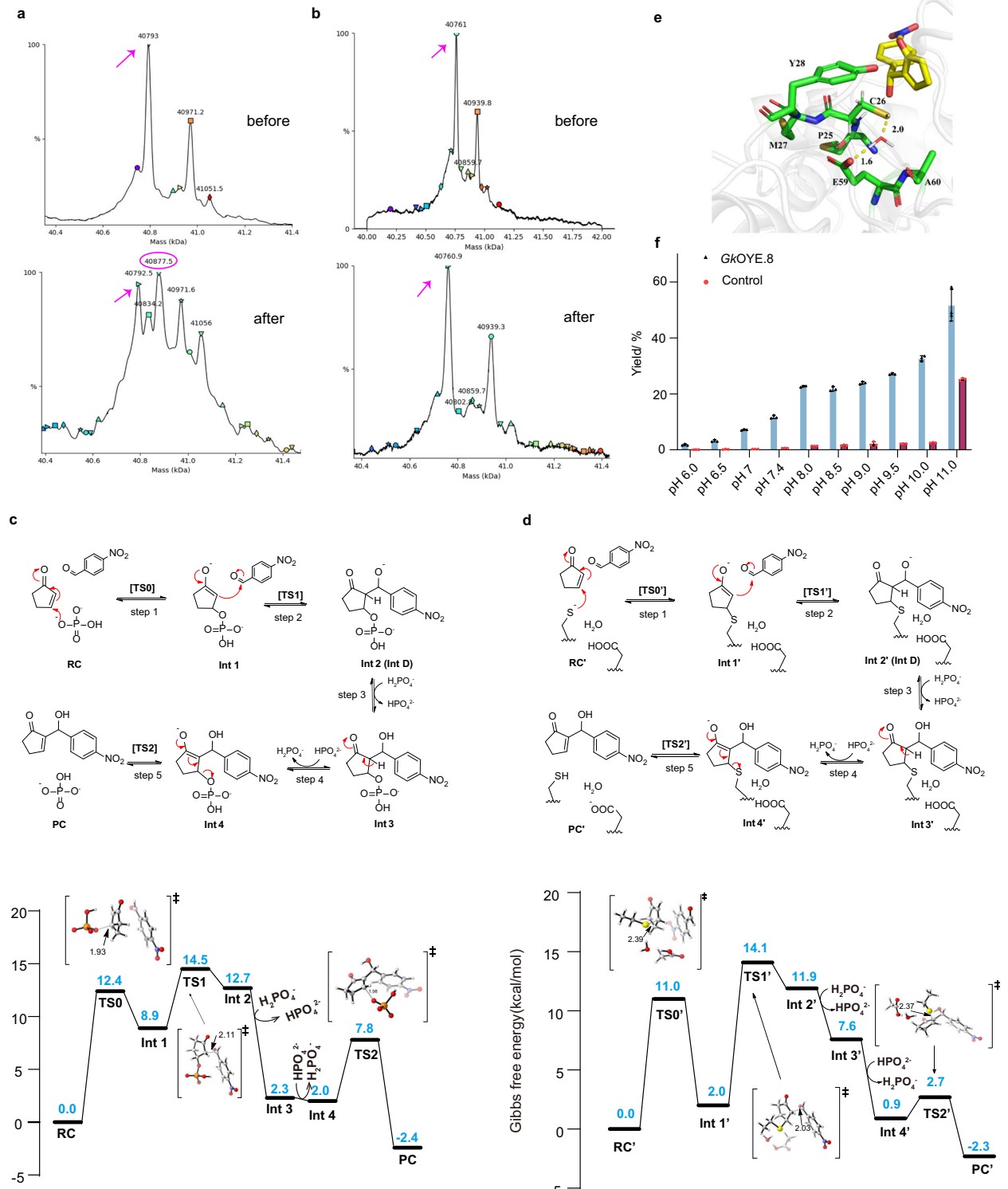

**Fig. 5 | The study on the nucleophilic catalytic residue of C26.**
**a** 2-Cyclopentenone labeling of *Gk*OYE.8 measured by ESI MS. **b** 2-Cyclopentenone labeling of *Gk*OYE.8[C26A] measured by ESI MS. Both (**a**) and (**b**) were labeled at 5 mM 2-cyclopentenone for 2 h. 2-cyclopentenone labels specifically on the *Gk*OYE.8 (peak position marked by red circle) but not on the *Gk*OYE.8[C26A]. **c** The proposed mechanism and Gibbs free energy profile of MBH reaction without enzyme calculated by DFT. The putative mechanism of MBH reaction consists of 5 steps[57,58]. The step 1: Michael addition. $HPO_4^{2-}$ as a nucleophile attacks substrate **1**[59]. The step 2: Aldol reaction. The step 3 and 4: Proton transfer. The step 5: Elimination. **d** The proposed mechanism and Gibbs free energy profile of MBH reaction with a

theozyme model calculated by DFT. The theozyme model includes substrate **1**, substrate **2**, water, methanethiol (substituting for residue C26) and acetic acid (substituting for residue E59). DFT-computed Gibbs free energies (in kcal/mol) at the CPCM(water)/B3LYP-D3/6-311 + + G(2d,p)//CPCM(water)/B3LYP-D3/6-31 + G(d) level of theory and transition-state structures (carbon: gray, hydrogen: white, oxygen: red, nitrogen: blue, sulfur: yellow, phosphorus: orange and distances are shown in Å). **e** The role of E59 site analyzed by the MD simulation. **f** Exploration the effect of pH on MBH reaction. *n* = 3 independent biological experiments. Data are presented as mean values ± SD. Source data are provided as a Source Data file.

catalytic performance. The first round of mutation aims to improve substrate binding affinity. MBH reaction is an unnatural reaction for OYE, and the substrates, especially substrate **2**, present a challenge in terms of effective binding within the pocket of *Gk*OYE.8. The kinetic

parameters of *Gk*OYE.8 revealed that the $K_m$ of substrate **2** (15.98 mM) was significantly higher than that of substrate **1** (4.28 mM) (Table 2), indicating a poor substrate binding affinity. To increase affinity toward substrate **2**, the strategy of introducing π-π interactions was considered due to the aromaticity of substrate **2**. Therefore, 14 different positions surrounding substrate **2** were selected for Trp scanning. The results were shown in Fig. 6a, mutant *Gk*OYE.9 (*Gk*OYE.8^G62W) exhibited the highest yield of **3** (48.4%) and (*R*)-preferred selectivity (82:18 e.r.), and the $K_m$ of *Gk*OYE.9 toward substrate **2** decreased by 3.5 mM, surprisingly, the $K_m$ value of substrate **1** also decreased by 2 mM, suggesting enhanced affinity of both substrates. Another effective Trp scanning mutant is *Gk*OYE.8^A104W displayed a yield and enantiomeric ratio of 41.4% and 73:37, respectively, however, combined A104W to *Gk*OYE.9 leads to 22.7 % decrease in **3** yield.

Based on *Gk*OYE.9, the second round of mutation entailed the application of saturation mutagenesis to residues (including P24, M25,

### Table 2 | Kinetic parameters of different mutants

| Enzyme | $K_m$ (1) [mM] | $k_{cat}$ (1) [h⁻¹] | $k_{cat}/K_m$ (1) [mM⁻¹h⁻¹] | $K_m$ (2) [mM] | $k_{cat}$ (2) [h⁻¹] | $k_{cat}/K_m$ (2) [mM⁻¹h⁻¹] |
|---|---|---|---|---|---|---|
| *Gk*OYE.8 | 4.28 | 0.64 | 0.15 | 15.98 | 1.65 | 0.10 |
| *Gk*OYE.9 | 2.00 | 0.65 | 0.33 | 12.44 | 1.59 | 0.13 |
| *Gk*OYE.11 | 1.70 | 0.7 | 0.41 | 11.22 | 1.72 | 0.15 |
| *Gk*OYE.12 | 4.50 | 0.60 | 0.13 | 17.28 | 1.52 | 0.09 |
| *Gk*OYE.13 | 1.91 | 0.68 | 0.36 | 12.10 | 1.66 | 0.14 |

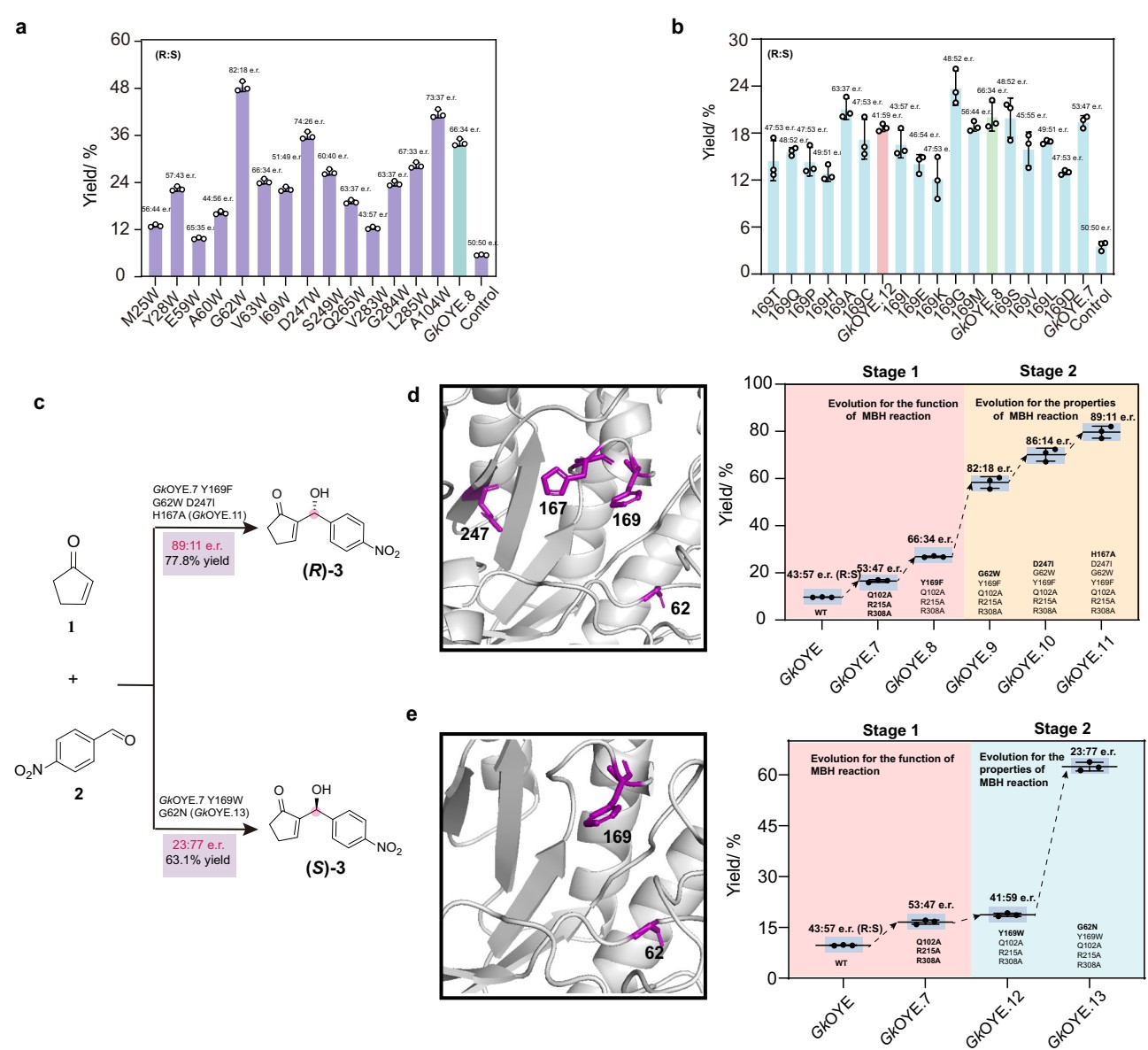

**Fig. 6 | Directed evolution of *Gk*OYE for MBH reaction. a** The results of Trp scanning. **b** The results of Y169 sites saturation mutation. **c** Evolved final variants *Gk*OYE.11 and *Gk*OYE.13 as orthogonal biocatalysts for enantioselective MBH reaction. **d** Yield and e.r. values of representative mutants during directed evolution for producing (*R*)-**3**. The reactions were carried out using 5 mM **1**, 1 mM **2**, and 100 μM purified protein in PBS (pH 10.0) with 3% methanol (MeOH) as a co-solvent. **e** Yield and e.r. values of representative mutants during directed evolution for producing (*S*)-**3**. *n* = 3 independent biological experiments. Data are presented as mean values ± SD. Source data are provided as a Source Data file.

I69, K109, E162, D247, H167, F169, V283, and A104; as depicted in Fig. 4e) that had a significant impact on yield. The results showed that the mutants GkOYE.9[H167A], GkOYE.9[A104C], GkOYE.9[D247I] exhibited yields of 62.4%, 65.4%, and 69.8%, with e.r. values of 82:18, 85:15, and 86:14, respectively. The mutant GkOYE.9[D247I] was denoted as GkOYE.10 and then introduced H167A and A104C, respectively. As shown in Fig. 6c, d, the optimal combination mutant GkOYE.11(GkOYE.10[H167A]) achieved 77.8% yield and 89:11 e.r. value. As shown in Table 2, by measuring the kinetic parameters of GkOYE.11, it was found that the $K_m$ value of substrates **1** and **2** further decreased by 0.3 mM and 1.22 mM, respectively, compared to that of GkOYE.9, and the $k_{cat}$ value of substrates **1** and **2** increased by 0.05 h$^{-1}$ and 0.13 h$^{-1}$, respectively. This indicates that the main reason for the improvement in catalytic efficiency was the enhancement of the binding affinity between substrate and protein. Moreover, this outcome indicates that mutations that enhance activity also result in heightened stereoselectivity.

In order to reverse the stereoselectivity of the GkOYE, we carried out the third round of mutation. The mutant GkOYE.8, which mutated Y169 to Phe, showing a significant improvement in stereoselectivity (Fig. 4c). Therefore, based on GkOYE.7, we conducted a saturation mutagenesis on Y169, as illustrated in Fig. 6b. When it mutated to Trp, the resulting mutant, GkOYE.12, gave 18.7% yield with (S)-preferred selectivity ((R)-**3**:(S)-**3** = 41:59 e.r.), in contrast to the stereoselectivity observed in GkOYE.8. This suggests that the residue Y169 likely plays a crucial role in influencing the configuration of the product, serving as a regulator of stereoselectivity. In Fig. 6d, it is clearly that introducing G62W into the mutant GkOYE.8 significantly improved stereoselectivity. Therefore, we performed saturation mutation at G62 based on GkOYE.12, and the results revealed that the mutant GkOYE.13 (GkOYE.12[G62N]), gave 61.3% yield with (S)-preferred selectivity (23:77 e.r.) (Fig. 6c, e). Based on GkOYE.13, we further performed saturation mutations on residues that had previously impacted yield (including P24, M25, G62, I69, K109, E162, D247, H167, F169, V283, and A104; as depicted in Fig. 4e), and the results showed a decrease in stereoselectivity for all mutants. Therefore, GkOYE.13 was obtained as the optimal mutant for (S)-preferred selectivity through directed evolution. As shown in Table 2, by measuring the kinetic parameters of GkOYE.13, it was found that the $K_m$ value of substrates **1** and **2** further decreased by 2.59 mM and 5.18 mM, respectively, compared to that of

GkOYE.12. This indicates that the main reason for the improvement in catalytic efficiency was the enhancement of the binding affinity between substrate and protein.

## Substrate scope of evolved MBHase

Substrate scope and limitations of the optimal mutant GkOYE.11 and GkOYE.13 were further explored under the optimal reaction conditions. Two unsaturated hydrocarbon substrates, a five-membered **1** and a six-membered **1a**, were subjected to testing alongside 10 different aromatic aldehyde coupling reagents featuring varying substituents (**2-2i**). This comprehensive analysis resulted in the synthesis of 11 structurally diverse MBH adducts, as illustrated in Fig. 7 and Supplementary Fig. 55. GkOYE.11 and GkOYE.13 accept a range of undecorated, mono-substituted and di-substituted aromatic aldehydes as substrates, although it is evident that para-substituted derivatives are preferred over meta-substituted derivatives. Among the five-membered ring para-substituted products, the yield exhibited a gradual decrease as the electron-withdrawing strength of the substituents decreased (-NO$_2$ > -CN > -Cl > -Br > -OMe). In contrast, the yield of **3c** is superior to that of **3b**. Remarkably, the most favorable performance, reaching 77.8%, 89:11 e.r. was obtained when using the coupling reagent featuring -NO$_2$ para-substituted benzaldehyde (**2**) catalyzed by GkOYE.11. Introduction of a strongly donating p-OMe group leads to a substantial reduction in activity (**3d**), the yield was only 0.2% and 0.3% catalyzed by GkOYE.11 and GkOYE.13, respectively. Interestingly, the undecorated product (**3 f**) exhibits lower yield compared to the para-electron-withdrawing substituent products (**3, 3a-3c**) but higher yield compared to the para-electron-donating substituent product (**3d**) (Supplementary Fig. 55). Surprisingly, it is notable that when GkOYE.11 and GkOYE.13 catalyze the undecorated aromatic aldehyde (**2e**) and the meta-substituted aromatic aldehydes (**2 f** and **2j**), there is a surprising stereoselective flip observed in the enantiomeric ratio of the product (Supplementary Fig. 55). This flip occurs despite the expected outcome based on the type of mutant used, indicating a complex and nuanced stereoselectivity pattern in these reactions. It can be seen that the stereoselectivity of the products varied based on the position and type of substituent. The aliphatic alkenes methyl acrylate, but-3-en-2-one and pent-1-en-3-one and the aliphatic aldehydes propionaldehyde, butyraldehyde, pentanal and

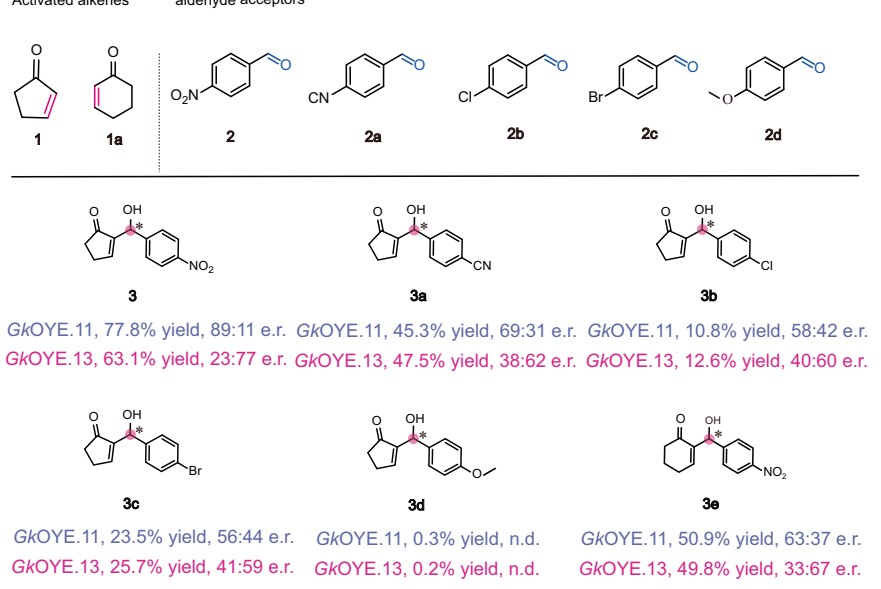

**Fig. 7 | Substrate scope of GkOYE.11 and GkOYE.13.** Biotransformations were performed using aldehyde (1 mM), activated alkene (5 mM) and catalyst (100 μM). The detailed information of results and determination methods for the yield and enantiomeric ratio of the MBH adducts were shown in Supplementary materials.

hexanal are not substrates of *Gk*OYE.11 and *Gk*OYE.13. The reason likely due to their increased affinity for aromatic substrates during directed evolution. Besides, mutants *Gk*OYE.11 and *Gk*OYE.13 showed modestly yield and enantiomeric ratio for six-membered ring substrates **1a**.

## Discussion

This study explores the promiscuous function of OYE to facilitate the MBH reaction. Initially, by considering the substrate similarity between the MBH reaction and its native reduction reaction, it was identified that *Gk*OYE demonstrates promiscuous functionality, catalyzing both reduction and MBH reactions. Afterward, an analysis of the structural basis of the catalytic residues in the *Gk*OYE-catalyzed MBH reaction was conducted, revealing that the catalytic residues responsible for the MBH reaction differ from those involved in its native reduction reaction. Moreover, employing mechanism and structural based protein engineering, the native reduction function of *Gk*OYE was inhibited, concurrently amplifying the C-C bond formation function, resulting in the *S*-selective mutant *Gk*OYE.13 and the *R*-selective mutant *Gk*OYE.11, showing enhanced catalytic efficiency and complementary stereoselectivity.

Native OYEs has been confirmed to possess promiscuous function in catalyzing the MBH reaction. OYEs belong to a class of ene-reductases, primarily used for catalyzing the reduction of $C = C$ bonds. In earlier studies, the function promiscuity of ene-reductases was mainly associated with the reduction of $C = X$ bonds, including promiscuously capabilities for reducing $C = N$[34,35], $N = O$, and $O = O$ bonds[36]. Currently, research on the function promiscuity of ene-reductases has mostly focused on the field of photocatalysis for C-C bond formation, such as radical cyclization reactions[11] and intermolecular radical alkene alkylation reactions[37]. Unexpectedly, based on substrate similarity, this study discovered the new function of ene-reductases in catalyzing the MBH reaction. In comparison, this reaction demonstrates an unexpected promiscuous function of ene-reductases for C-C bond formation, which does not require cofactors or assistance from light. It highlights the rich catalytic versatility of ene-reductases, expanding the toolbox for enzymatic transformations.

This study has revealed that the catalytic residues responsible for the MBH reaction is entirely different from that involved in its native reduction reaction. Firstly, in MBH reaction, C26 serves as a nucleophilic catalyst, while in natural reduction reactions, C26 might function as a redox sensor, controlling the oxidation-reduction potential of flavin based on the presence of substrates[28]. Secondly, in MBH reaction, E59 is responsible for deprotonating the thiol group on C26 through a water-mediated hydrogen bond network, enhancing the nucleophilicity of -S⁻, while the specific function of E59 in natural reduction reactions has not been reported. Additionally, the key catalytic residue Y169, which serves the role of providing H⁺ in native reduction reactions, transforms into a knob controlling stereoselectivity in the MBH reaction. Mutating Y169 to Phe tends to favor the generation of (*R*)-MBH adducts, while mutated to Trp tends to favor (*S*)-MBH adducts. The last key residue is H167, and initially, it was anticipated that H167 would play a role in stabilizing the oxyanion intermediate in the MBH reaction like in native reduction reactions. However, when mutated H167 to Ala, the MBH reaction yield is enhanced, suggesting that H167 does not participate in stabilizing the oxyanion intermediate and may have disadvantageous on MBH reaction. According to reported study, the efficient occurrence of MBH reaction relies on the concerted action of nucleophilic catalytic residues (such as His, Cys) and residues stabilizing the oxyanion intermediate (such as Arg, His)[32,38,39]. Therefore, the most likely reason for the low catalytic efficiency of *Gk*OYE mutants in catalyzing the MBH reaction in this study is that the H167 residue, originally designed to stabilize the oxyanion, did not function as expected.

This study represents a typical case of converting the low promiscuous MBH activity of enzymes into the main activity, resulting in a set of enantiocomplementary enzymes for the asymmetric synthesis of product **3**. Previous studies have reported issues with low catalytic efficiency[40], poor selectivity[22] and single chirality[23]. In comparison, this study achieved a reversal of functional selectivity by attenuating the native reduction function of *Gk*OYE and enhancing the C-C bond formation function. Subsequent protein engineering continuously enhanced the activity and selectivity of the MBH reaction, ultimately obtaining a set of enantiocomplementary mutants, *Gk*OYE.11 and *Gk*OYE.13, for generating (*R*)-**3** (77.8% yield and 89:11 e.r.) and (*S*)-**3** (63.1% yield and 23: 77 e.r.), respectively. Unfortunately, it is crucial to acknowledge that the catalytic activity of the *Gk*OYE mutants still falls short when compared to BH32.14. Further reshaping of their catalytic pockets or screening for other OYEs is necessary to enhance their efficiency in catalyzing the MBH reaction.

## Methods

### General

Commercial reagents, standards, and solvents were purchased from Sigma Aldrich (Shanghai, China), Meryer Chemicals (Shanghai, China), Aladdin Reagents (Shanghai, China), J&K Chemicals (Beijing, China), and TCI Chemicals (Shanghai, China), and used without further purification. All codon-optimized genes and primers used in this study were listed in Supplementary Date 1.

### Fermentation medium and conditions

A single colony of recombinant *E. coli* strain was cultivated overnight (10–12 h, 37 °C) in LB medium (10 g·L⁻¹ peptone, 5 g·L⁻¹ yeast extract, and 10 g·L⁻¹ NaCl; pH 7.0) with appropriate antibiotics (50 µg·mL⁻¹ kanamycin) and used as the inoculum (1%). The culture was then transferred into 200 mL 2×YT medium (10 g·L⁻¹ yeast extract, 16 g·L⁻¹ peptone, 5 g·L⁻¹ NaCl; pH 7.0) containing appropriate antibiotics in a 500 ml flask. When the OD₆₀₀ of the culture broth reached 0.6-0.8, isopropyl β-D-1-thiogalactopyranoside (IPTG) was added to a final concentration of 0.2 mM to induce gene expression. The cells were inducted at 16 °C for 16 h and collected by centrifugation (8000 × *g*, 10 min).

### Enzyme purification

The cell pellet was resuspended in 30 ml washing buffer (50 mM Na₂HPO₄, 150 mM NaCl, pH 8.0) containing a protease inhibitor and subsequently lysed by high pressure homogenizer. The lysed cells were incubated with DNase I (final concentration 0.1 mg·ml⁻¹) for 30 min at 4 °C. After removing cell debris by centrifugation (10,000 r.p.m., 30 min, 4 °C). The cleared lysate was loaded on a Strep-Tactin column (Strep-Tactin Superflow high capacity), incubated for 60 min at 4 °C, and the protein was purified according to the manufacturer's guidelines. Proteins were desalted using 10DG desalting columns (Bio-Rad) with PBS pH 10.0 and analysed by SDS−PAGE. When it is necessary to perform protein crystallization experiments, the subsequent experiments were performed on an ÄKTA pure system (GE Healthcare) with a HisTrap HP column (5 ml, GE Healthcare). Protein concentration of purified enzyme was measured by detecting absorbance at 280 nm using a NanoPhotometer N50 spectrophotometer and considering the calculated extinction coefficients with the ExPASy ProtParam Tool.3. All purification operations were conducted at 4°C when necessary. The uncropped and unprocessed gels scans were provided in the Source Data and Supplementary Information.

### HPLC analysis

Analysis of the concentration of adduct **3** was carried out using Waters Alliance e2695 HPLC (Waters Co., USA) with UV detector at 269 nm. Analysis of the concentration of adduct **3a-3g** was carried out using Waters Alliance e2695 HPLC (Waters Co., USA) with UV detector at 254 nm. Column: Agilent SB-C18 (150×4.6 mm, 5 µm), mobile phase: acetonitrile: H₂O (0.7% HAc) = 30:70, flow rate: 1 mL·min⁻

[1], temperature: 25 °C. Peaks were assigned by comparison to chemically synthesized standard.

The separation and purification of adduct **3** was carried out using preparative high performance liquid chromatography Waters 2545 Binary Gradient Module (Waters Co., USA) with a Waters 2767 Sample Manager, and the UV detector at 269 nm. Column: CST Daiso C18 (250 × 20 mm, 10 μm, 12 nm), mobile phase: acetonitrile: $H_2O$ (0.7% HAc) = 30:70, flow rate: 10 mL·min$^{-1}$, temperature: 25 °C.

The stereoselectivity of MBH adduct **3** were determined using Waters Alliance e2695 HPLC (Waters Co., USA) with UV detector at 269 nm. The stereoselectivity of MBH adduct **3a-3g** were determined using Waters Alliance e2695 HPLC (Waters Co., USA) with UV detector at 254 nm. Column: DAICEL CHIRALPAK IC (250 × 4.6 mm, 5 μm), flow rate: 1 mL·min$^{-1}$, temperature: 25 °C, mobile phase: n-hexane/isopropanol=90/10 (v/v). Peaks were assigned by comparison to the results of BH32.14[23].

## GC and GC-MS analysis

The concentrations of **3** and **4** were determined by using a GC (GC-2030, SHIMADZU, Japan) with a column (SH-5, 15 m × 0.25 mm × 0.1 μm; SHIMADZU). The GC analysis conditions were as follows: injector 310 °C, and oven 45 °C, hold 2 min, 10 °C min$^{-1}$ to 280 °C, hold 10 min.

The identification of adducts (**3, 3a-3g**) were determined by using a GC-MS (Trace1310-TSQ8000, Thermo Scientific, USA) with a column (TG-5 30 m×0.25 mm×0.25 μm). The GC analysis conditions were as follows: injector 270 °C, ion source temperature 300 °C, and oven 80 °C, 15 °C min$^{-1}$ to 280 °C, hold 8 min. Sample processing: 300 μL reaction samples, extraction experiments were performed after the addition of an equal volume of ethyl acetate, centrifuged at 14,000 × $g$ for 10 min, filtered by 0.22 μm filter membrane, and then determined by GC-MS detection (according to the situation or ethyl acetate appropriate dilution), for determining the content of the product. Three times of biological replicates.

## Activity assay

The activity of MBHase was measured by HPLC. The assay mixture contained 300 μL PBS buffer (pH 10.0), including 1 mM **2**, 5 mM **1** and 30 μM purified protein. Reactions were conducted in triplicate and incubated at 25 °C and 220 rpm for 10 h and started by addition of the enzyme solution. The reaction was terminated with equal volume of 1 M HCl, centrifuged at 14,000 × $g$ for 10 min, filtered by 0.22 μm filter membrane, and then determined by HPLC. One unit of specific activity is defined as the amount of adduct **3** (in nmol) that can be produced by one unit of enzyme in per minute under standard conditions.

## Determination methods for the yield

The molar concentration of the product was quantified using the external standard method. Subsequently, the yield of different products were determined using

$$\text{yield} = \frac{C(\text{Product}) * \text{dilution factor}}{C(\text{substrate})} * 100\%, \quad (1)$$
$$C: \text{molar concentration.}$$

## Kinetic characterization

Initial velocity ($V_0$) versus [4-nitrobenzaldehyde] kinetic data were measured using strep-tagged purified enzyme (30 μM), a fixed concentration of **1** (5 mM) and varying concentrations of 2 (0.5–35 mM). Reactions were performed in PBS pH 10.0 with 3% methanol and were incubated at 25 °C with shaking (220 r.p.m.) for 12 h. $V_0$ versus [2-cyclohexen-1-one] kinetic data were measured using a fixed concentration of **2** (4 mM) and varying concentrations of **1** (0.5–10 mM) using the enzyme concentrations and buffer conditions described above. Samples were quenched with 1 vol. of 1 M HCl and analyzed by HPLC. The $K_m$ and $k_{cat}$ values were calculated by nonlinear regression according to the Michaelis–Menten equation using GraphPad Prism software.

## Mass spectrometry

Purified protein samples were labeled at 5 mM 2-cyclopentenone for 2 h and buffer-exchanged into PBS (pH 10.0) using a 10 k MWCO Vivaspin unit (Sartorius) and diluted to a final concentration of 0.2 μM and then add 0.2% acetic acid. MS was performed using LTQ-Orbitrap Velos system, mass range: 500–1500, max inject time: 10 ms, resolution: 30000, sheath gas flow rate: 30, aux gas flow rate: 5, sweep gas flowrate: 1, capliary temp: 275°C, S-Lens RF Level: 69%, flow rate: 2 μL·min$^{-1}$, record: 10 min. The resulting multiply charged spectrum was analyzed and deconvoluted using Unidec software. The total number of sample and control are both one, three times of biological replicates, the control are $Gk$OYE.8$^{C26A}$.

## LC-MS

detection was performed on a Sciex LC coupled to a triple quadrupole trap mass spectrometer (QTRAP 5500, AB Sciex, USA) (Q-TRAP-MS) with electrospray ionization (ESI) in negative mode. The HPLC analysis was performed on the BEH C18 column (2.5 μm, 2.1 × 50 mm; Waters) at a temperature of 40 °C. The mobile phase was a mixture of two solvents: A- water (0.1% FA) and B- acetonitrile. The optimized linear gradient system was as follows: 0 min, 95 % A; 0–1 min to 95% A; 1–5 min, to 5% A; 5–7 min, 5% A; 7–10 min to 95% A. The autosampler was set to 4 °C. The injection volume was 2 μL, and the flow rate was 300 μL/min. The injection needle was washed after each injection with acetonitrile. The instrument parameters were as follows: ion spray voltage (IS): 5500 V; temperature: 550 °C; nebulizer gas (GS1): 60 psi; turbo gas (GS2): 60 psi; curtain gas (CUR): 35 psi; and collision gas (CAD): medium. Instrument control and data integration were performed using Analyst® Software Version 1.6.2. Sample processing: Protein samples were boiled at 100 °C for 10 min and centrifuged at 14,000 × $g$ for 10 min, the supernatant was collected and filtered through 0.22 μm filter membrane and then determined by LC-MS detection (according to the situation or ethyl acetate appropriate dilution). The total number of sample and control are 1 and 2, respectively. three times of biological replicates, the controls are the protein of wild type of $Gk$OYE and the commercial reagent FMN.

## Biotransformation procedures. For the reduction reaction catalyzing by $Gk$OYE

Reactions were performed in 300 μL PBS buffer (pH 7.4) with 5 mM **1**, 1 mM NADPH, 100 μM purified protein and 1% methanol as cosolvent. Reactions were incubated at 25 °C and 220 rpm for 8 h. The solution was extracted with the same volume ethyl acetate (EtOAc). The resulting solution was dried by $Na_2SO_4$ and filtered through 0.22 μm membrane filters. Then the yield determined via GC.

## For the MBH reaction catalyzing by $Gk$OYE and mutants

For the product **3, 3a-3g**, reactions were performed in 300 μL PBS buffer (pH 10.0) with 5 mM **1,1a**, 1 mM **2, 2a-2i**, 100 μM purified protein and 3% methanol as cosolvent. Reactions were incubated at 25 °C and 220 rpm for 40 h. The reaction was terminated with equal volume of 1 M HCl, centrifuged at 14,000 × $g$ for 10 min, filtered by 0.22 μm filter membrane, and then determined by HPLC. For the detection of stereoselectivity of MBH adducts. The solution was extracted with the same volume ethyl acetate (EtOAc). The resulting solution was dried by $Na_2SO_4$ and filtered through 0.22 μm membrane filters. Then the stereoselectivity was determined via HPLC with chiral column.

All the experiments were carries out at least in duplicate. The above products were further identified by nuclear magnetic resonance (NMR) analysis and shown in Supplementary Fig. 13–22. The representative HPLC chromatograms are shown in Supplementary

Fig. 23–44. The representative GC-MS chromatograms are shown in Supplementary Fig. 45–54.

## Preparation of racemic products standards

2-(hydroxy(4-nitrophenyl)methyl)cyclopent-2-en-1-one (3) was separation and purification using preparative high performance liquid chromatography. The instrument and method use Waters 2545 Binary Gradient Module (Waters Co., USA) with a Waters 2767 Sample Manager, and the UV detector at 269 nm. Column: CST Daiso C18 (250×20 mm, 10 μm, 12 nm), mobile phase: acetonitrile: $H_2O$ (0.7% HAc) = 30:70, flow rate: 10 mL·min$^{-1}$, temperature: 25 °C. The spectral data are consistent with literature values[41]. **¹H NMR** (600 MHz, Chloroform-d) δ 8.19 (d, J = 8.4 Hz, 2H), 7.57 (d, J = 8.4 Hz, 2H), 7.31 (s, 1H), 5.66 (s, 1H), 3.74 (s, 1H), 2.66–2.59 (m, 2H), 2.52 - 2.42 (m, 2H). **¹³C NMR** (151 MHz, Chloroform-d) δ 209.45, 160.07, 148.68, 147.58, 146.85, 127.21, 123.83, 69.02, 35.26, 26.96. **MS (EI):** m/z calcd for $C_{12}H_{11}NO_4$: 233.07; found: 233.1.

General procedure for the preparation of racemic product standards (**3a**-**3g**) was refer to the literature[23]. Arylaldehyde (1.5 mmol, 1.5 equiv.), cyclic enone (3 mmol, 3.0 equiv.), KOH (1 mmol, 1.0 equiv.) and imidazole (1 mmol, 1.0 equiv.) were stirred in 1 M $NaHCO_3$ (2 ml) and THF (2 ml) for 48 h at room temperature. The reaction was acidified with 1 M HCl and extracted with EA (3 × 50 ml). The organic layer was dried over $MgSO_4$ filtered, and the solvent was removed in vacuo to give the crude product. The crude product was purified by silica gel chromatography (PE:EA = 2:1).

**4-(hydroxy(5-oxocyclopent-1-en-1-yl)methyl)benzonitrile (3a).** The crude product was purified by flash chromatography (PE:EA = 2:1) to give the product (101.7 mg, 31.8%).The spectral data are consistent with literature values[42]. **¹H NMR** (600 MHz, Chloroform-d) δ 7.63 (d, J = 8.4 Hz, 2H), 7.51 (d, J = 8.4 Hz, 2H), 7.30 (t, J = 2.7 Hz, 1H), 5.60 (s, 1H), 3.70 (s, 1H), 2.64–2.56 (m, 2H), 2.49 - 2.41 (m, 2H). **¹³C NMR** (151 MHz, Chloroform-d) δ 209.38, 159.90, 146.93, 146.78, 132.41, 127.09, 118.80, 111.64, 69.17, 35.25, 26.91. **MS (EI):** m/z calcd for $C_{13}H_{11}NO_2$: 213.08; found: 213.1.

**2-((4-chlorophenyl)(hydroxy)methyl)cyclopent-2-en-1-one (3b).** The crude product was purified by flash chromatography (PE:EA = 2:1) to give the product (75.5 mg, 22.7%). The spectral data are consistent with literature values[43]. **¹H NMR** (600 MHz, Chloroform-d) δ 7.31 (s, 4H), 7.26 (t, J = 2.7 Hz, 1H), 5.51 (s, 1H), 2.62 - 2.55 (m, 2H), 2.49 - 2.39 (m, 2H). **¹³C NMR** (151 MHz, Chloroform-d) δ 209.66, 159.63, 147.54, 139.97, 133.68, 128.74, 127.86, 69.28, 35.33, 26.80. **MS (EI):** m/z calcd for $C_{12}H_{11}ClO_2$: 222.04; found: 222.07.

**2-((4-bromophenyl)(hydroxy)methyl)cyclopent-2-en-1-one (3c).** The crude product was purified by flash chromatography (PE:EA = 2:1) to give the product (104.8 mg, 26.2%). **¹H NMR** (400 MHz, Chloroform-d) δ 7.44 – 7.41 (m, 2H), 7.28 – 7.27 (m, 1H), 7.24 – 7.20 (m, 2H), 5.46 – 5.44 (m, 1H), 3.84 (d, J = 4.0 Hz, 1H), 2.57 – 2.54 (m, 2H), 2.41 – 2.38 (m, 2H). **¹³C NMR** (101 MHz, Chloroform-d) δ 209.3, 159.5, 147.3, 140.4, 131.4, 128.0, 121.5, 68.8, 35.1, 26.6. **MS (EI):** m/z calcd for $C_{12}H_{11}BrO_2$: 267.12; found: 266.97.

**2-(hydroxy(4-methoxyphenyl)methyl)cyclopent-2-en-1-one (3d).** The crude product was purified by flash chromatography (PE:EA = 2:1) to give the product (37.5 mg, 11.5%). The spectral data are consistent with literature values[43]. **¹H NMR** (600 MHz, Chloroform-d) δ 7.30 – 7.27 (m, 3H), 6.88 – 6.85 (m, 2H), 5.48 (s, 1H), 3.78 (s, 3H), 3.48 (s, 1H), 2.59 – 2.56 (m, 2H), 2.45 – 2.42 (m, 2H). **¹³C NMR** (151 MHz, Chloroform-d) δ 209.56, 159.2, 159.1, 147.9, 133.5, 127.6, 113.8, 69.4, 55.2, 35.2, 26.5. **MS (EI):** m/z calcd for $C_{13}H_{14}O_3$: 218.09; found: 218.11.

**2-(hydroxy(4-nitrophenyl)methyl)cyclohex-2-en-1-one (3e).** The crude product was purified by flash chromatography (PE:EA = 2:1) to give the product (47.1 mg, 12.7%). The spectral data are consistent with literature values[43]. **¹H NMR** (600 MHz, Chloroform-d) δ 8.15 (d, J = 8.8 Hz, 2H), 7.52 (d, J = 8.8 Hz, 2H), 6.83 (t, J = 4.2 Hz, 1H), 5.59 (d, J = 5.6 Hz, 1H), 3.67 (d, J = 5.8 Hz, 1H), 2.44 – 2.40 (m, 4H), 2.01–1.97 (m, 2H). **¹³C NMR** (151 MHz, Chloroform-d) δ 200.0, 149.4, 148.1, 147.1, 140.1, 127.1, 123.4, 71.7, 38.3, 25.7, 22.3. **MS (EI):** m/z calcd for $C_{13}H_{13}NO_4$: 247.08; found: 247.12.

**2-(hydroxy(phenyl)methyl)cyclopent-2-en-1-one (3f).** The crude product was purified by flash chromatography (PE:EA = 2:1) to give the product (175 mg, 62.0%). The spectral data are consistent with literature values[43]. **¹H NMR** (400 MHz, Chloroform-d) δ 7.40–7.32 (m, 4H), 7.31–7.25 (m, 2H), 5.55 (s, 1H), 3.58 (brs, 1H), 2.59–2.56 (m, 2H), 2.45–2.42 (m, 2H). **¹³C NMR** (101 MHz, Chloroform-d) δ 209.5, 159.3, 147.7, 141.3, 128.4, 127.8, 126.3, 69.7, 35.2, 26.6. **MS (EI):** m/z calcd for $C_{12}H_{12}O_2$: 188.23; found: 188.1.

**2-(hydroxy(3-nitrophenyl)methyl)cyclopent-2-en-1-one (3j).** The crude product was purified by flash chromatography (PE:EA = 2:1) to give the product (136.8 mg, 39.1%).The spectral data are consistent with literature values[43]. **¹H NMR** (400 MHz, Chloroform-d) δ 8.24 (t, J = 2.0 Hz, 1H), 8.13 (ddd, J = 8.2, 2.4, 1.0 Hz, 1H), 7.74 (dt, J = 7.8, 1.3 Hz, 1H), 7.52 (t, J = 7.9 Hz, 1H), 7.35 (td, J = 2.7, 1.2 Hz, 1H), 5.65 (dd, J = 4.3, 1.8 Hz, 1H), 3.78 (d, J = 4.4 Hz, 1H), 2.65 – 2.62 (m, 2H), 2.48 – 2.45 (m, 2H). **¹³C NMR** (101 MHz, Chloroform-d) δ 209.23, 159.9, 148.3, 146.7, 143.5, 132.5, 129.4, 122.7, 121.2, 68.9, 35.1, 26.8. **MS (EI):** m/z calcd for $C_{12}H_{11}NO_4$: 233.22; found: 233.07.

**3-(hydroxy(5-oxocyclopent-1-en-1-yl)methyl)benzonitrile (3 h).** The crude product was purified by flash chromatography (PE:EA = 2:1) to give the product (100.4 mg, 31.4%). **¹H NMR** (400 MHz, Chloroform-d) δ 7.68 (t, J = 1.7 Hz, 1H), 7.63 (dt, J = 7.8, 1.6 Hz, 1H), 7.56 (dt, J = 7.7, 1.5 Hz, 1H), 7.45 (t, J = 7.7 Hz, 1H), 7.32 (td, J = 2.7, 1.2 Hz, 1H), 5.57 (dd, J = 4.4, 1.8 Hz, 1H), 3.73 (d, J = 4.4 Hz, 1H), 2.64–2.61 (m, 2H), 2.47 – 2.44 (m, 2H). **¹³C NMR** (101 MHz, Chloroform-d) δ 209.2, 159.7, 146.9, 142.9, 131.4, 130.8, 129.9, 129.2, 118.6, 112.5, 68.8, 35.1, 26.7. **MS (EI):** m/z calcd for $C_{13}H_{11}NO_2$: 213.24; found: 213.1.

**2-((2-fluoro-4-nitrophenyl)(hydroxy)methyl)cyclopent-2-en-1-one (3i).** The crude product was purified by flash chromatography (PE:EA = 2:1) to give the product (57.6 mg, 15.3%). **¹H NMR** (600 MHz, Chloroform-d) δ 8.08 (dd, J = 8.6, 2.1 Hz, 1H), 7.91 (dd, J = 9.7, 2.2 Hz, 1H), 7.80 (dd, J = 8.5, 7.2 Hz, 1H), 7.28 – 7.27 (m, 1H), 5.90 (d, J = 4.2 Hz, 1H), 4.13 (d, J = 5.0 Hz, 1H), 2.69 – 2.58 (m, 2H), 2.53 – 2.45 (m, 2H). **¹³C NMR** (151 MHz, Chloroform-d) δ 209.7, 160.1, 158.8 (d, J = 251.7 Hz), 148.1 (d, J = 8.8 Hz), 144.6, 136.0 (d, J = 13.2 Hz), 128.7 (d, J = 4.3 Hz), 119.6 (d, J = 3.3 Hz), 111.1 (d, J = 27.1 Hz), 63.9 (d, J = 3.2 Hz), 35.1, 26.8. **¹⁹F NMR** (565 MHz, Chloroform-d) δ -113.74. **MS (EI):** m/z calcd for $C_{12}H_{10}FNO_4$: 251.21; found: 251.0.

**2-((3-fluoro-4-nitrophenyl)(hydroxy)methyl)cyclopent-2-en-1-one (3 g).** The crude product was purified by flash chromatography (PE:EA = 2:1) to give the product (74.6 mg, 19.8%). **¹H NMR** (600 MHz, Chloroform-d) δ 8.04 – 8.01(m, 1H), 7.39 – 7.37(m, 2H), 7.31 (dd, J = 8.5, 1.6 Hz, 1H), 5.62 – 5.61 (m, 1H), 3.77 (d, J = 4.4 Hz, 1H), 2.66–2.64 (m, 2H), 2.48 – 2.46 (m, 2H). **¹³C NMR** (151 MHz, Chloroform-d) δ 209.1, 160.2, 155.6 (d, J = 265.0 Hz), 150.5 (d, J = 7.6 Hz), 146.2, 136.4 (d, J = 7.4 Hz), 126.2, 122.1 (d, J = 3.4 Hz), 116.0 (d, J = 21.8 Hz), 68.3, 35.1, 26.8. **¹⁹F NMR** (565 MHz, Chloroform-d) δ -116.54. **MS (EI):** m/z calcd for $C_{12}H_{10}FNO_4$: 251.21; found: 251.0.

## Conservation analysis of the residues

Using The online Consurf Server (https://consurf.tau.ac.il/) and protein structure display software Pymol to analyze the residues conservation of *Gk*OYE.

## Measurement the volume of catalytic pocket

The volume of catalytic pockets of different mutants was measured using DoSiteScorer in the on-line software PROTEINS PLUS[44,45].

## Crystallization, refinement and model building

All initial conditions of crystallization were screened using the sitting drop vapor diffusion method with Hampton Research Crystal Screen Kits. The crystal of the *apo-Gk*OYE.8 was obtained by optimization after 2 days at 20 °C in hanging-drop plates, under the conditions of mixing 0.8 µL protein solution (15 mg·mL⁻¹) and an equal volume of reservoir solution (0.1 M glycine pH 9.0, 25% (w/v) polyethylene glycol 2000, 15% (w/v) glycerol). The crystal of the *apo-Gk*OYE.8 is shown in Supplementary Fig. 6. The crystals were cryoprotected by transient soaking in reservoir solutions with an additional 20-25% glycerol, and the crystals were flash cooled in liquid nitrogen at 100 K for data collection. X-ray diffraction data were collected on a Bruker D8 QUEST diffractometer (Karlsruhe, Germany), and the data sets were indexed, integrated and merged using XDS. The crystal structure of the *apo-Gk*OYE.8 was solved by the single anomalous scattering method using a crystal that was diffracted to a 3.11 Å resolution. The structure of the *apo-Gk*OYE.8 was solved using the molecular replacement method with a CCP4 automatic. The protein structure of 3gr7 was used as a searching model. Structural refinement was achieved using the Coot[46] and Refmac5[47] programs. The data collection and refinement statistics of the *apo-Gk*OYE.8 crystal structure are listed in Supplementary Table 2 and have been deposited in the PDB under accession code 8X0J. Structural figures were prepared using PyMOL v2.3.3 (Schrödinger, LLC, New York, USA).

## The secondary structure of key mutants detected by circular dichroism

**Preparation of protein**. The purified target protein is tested by SDS-PAGE for protein purity. When the target protein content reaches 95% or above (the higher the protein purity, the more reliable the experimental results), the purified protein can be used for circular dichroism spectroscopy. Circular dichroic determination using JASCO-1700 circular dichroic spectrometer made in Japan, the spectra of 190–240 nm in the far ultraviolet were determined at room temperature. The concentration of the sample was 0.2 mg·mL⁻¹ and the radius of the sample cup was 0.1 cm. The resolution is 0.5 nm, the bandwidth is 0.5 nm, the sensitivity is 50 mdeg, and the speed is 0.8 nm·min⁻¹. 0.1*PBS buffer was used for the background solution. In the process of sample determination, 0.1*PBS buffer was first measured as the background, and then the mutant protein after buffer replacement. The resulting spectrogram data was analyzed by Dichroweb.

## Library construction and screening

The primer sequences used to generate DNA libraries are shown in Supplementary Data 1. Site-directed saturation mutation was performed by PCR using mutagenic primers and plasmid pET28a-*Gk*OYE.8 as template according to the manufacturer's instructions of Quick-Change (Stratagene). The *Dpn*I-digested PCR product of 3 µl was used to transform 80 µl of *E. coli* BL21(DE3) chemically competent cells and colonies after transformation were incubated for DNA sequencing until all the designed mutants were obtained. A single colony of recombinant *E. coli* strain was cultivated overnight (10–12 h, 37 °C) in LB medium with 50 µg·mL⁻¹ kanamycin and used as the inoculum (1%). The culture was then transferred into 200 mL 2×YT medium containing appropriate antibiotics in a 500 ml flask. When the $OD_{600}$ of the culture broth reached 0.6-0.8, isopropyl β-D-1-thiogalactopyranoside

(IPTG) was added to a final concentration of 0.2 mM to induce gene expression. The cells were inducted at 16 °C for 16 h and collected by centrifugation (8000 × *g*, 10 min). The cell pellet was resuspended in 30 ml washing buffer containing a protease inhibitor and subsequently lysed by high pressure homogenizer. The lysed cells were incubated with DNase I (final concentration 0.1 mg·ml⁻¹) for 30 min at 4 °C. After removing cell debris by centrifugation (10,000 r.p.m., 30 min, 4 °C). The cleared lysate was loaded on a Strep-Tactin column (Strep-Tactin Superflow high capacity), incubated for 60 min at 4 °C, and the protein was purified according to the manufacturer's guidelines. Proteins were desalted using 10DG desalting columns (Bio-Rad) with PBS pH 10.0 and analysed by SDS–PAGE. Protein concentration of purified enzyme was measured by detecting absorbance at 280 nm using a NanoPhotometer N50 spectrophotometer and considering the calculated extinction coefficients with the ExPASy ProtParam Tool.3. All purification operations were conducted at 4°C when necessary. Then the pure protein of different mutants were carried out for conversion reaction. The reactions were performed in 300 µL PBS buffer (pH 10.0) with 5 mM **1**, 1 mM **2**, 100 µM purified protein and 3% methanol as cosolvent. Reactions were incubated at 25 °C and 220 rpm for 40 h. The reaction was terminated with equal volume of 1 M HCl, centrifuged at 14,000 × *g* for 10 min, filtered by 0.22 µm filter membrane, and then determined by HPLC.

## Initial structural preparation for computational studies

The initial structure of *Gk*OYE.8 were obtained by X-ray diffraction. The protonation states of the charged residues were determined at a constant pH of 10.0, based on *pKa* calculations via the H⁺⁺ server (http://biophysics.cs.vt.edu/H⁺⁺) and the consideration of the local hydrogen bonding network. In the *Gk*OYE.7 models, residues His41, 44, 81, 95 and 167 were set as HIE, and residues His164, and 222 were set as HID. In the model, all Asp and Glu residues were deprotonated, while the Lys and Arg residues were protonated. The bond and angle force constants were determined using the Seminario method[48], and point charge parameters for electrostatic potentials were determined using the ChgModB method. Each model was neutralized by the addition of Na⁺ ions and solvation in a truncated octahedral TIP3P water box with a buffer distance of 10 Å on each side. (*R*)-**3** was optimized at the B3LYP-D3/6-31 G(d,p) level by using Gaussian 16, the partial charge of these ligands was fitted with HF/6-31 G(d) calculations and the restrained electrostatic potential protocol[49] implemented by the Antechamber module in the Amber 18 package. The force field parameters for these ligands were adapted from the standard general Amber force field 2.0 (gaff2)[50] parameters, while the standard Amber19SB force field was applied to describe the protein.

## Molecular docking

To dock the (*R*)-**3** to the active sites of the *Gk*OYE.8, 50000 uniformly distributed snapshots from the 100-ns MD simulation (with time intervals of 2 ps) were selected and divided into ten groups using a hierarchical agglomerative (bottom-up) approach. The optimized substrate (*R*)-**3** were docked to the active site of one representative group snapshot to mimic the ligand-protein complex. Molecular docking was performed with the Lamarckian genetic algorithm local search method using AutoDock Vina[51]. The docking approach was used for a rigid receptor conformation, while all rotatable torsion bonds of (*R*)-**3** were left free. A grid box was centered near the residues 26, 164, 167 and 169, and its size was set at $20 \times 20 \times 20$ Å with a spacing of 0.375 Å. A total of 500 independent docking runs were performed with a maximum energy evaluation of $2.5 \times 10^7$. The 500 docked conformations obtained were clustered with an RMSD of 2.0 Å and ranked using an energy-based scoring function. The possible catalytically active binding modes were selected as initial configurations to perform MD simulations of *Gk*OYE.8 in complex with (*R*)-**3**, according to the scoring function and reasonable conformation.

## MD simulations

All MD simulations were performed using the Amber 18 package software[52]. The MD pre-equilibrated *Gk*OYE.8, and possible catalytically active binding modes of (*R*)-**3** was used as initial conformations for MD simulations of the protein-ligand complexes. Each system was brought to equilibrium with a series of minimizations interspersed by short MD simulations, during which restraints on the heavy atoms of the protein backbone were gradually released (with force constants of 10, 2, 0.1, and 0 kcal [mol $Å^{-2}$]) and then slowly heated from 0 to 300 K for 50 ps. Finally, a standard unrestrained 100-ns MD simulation with periodic boundary conditions at 300 K and 1 atm was performed. The pressure was maintained at 1 atm and coupled with isotropic position scaling. The temperature was maintained at 300 K using the Berendsen thermostat. Long-range electrostatic interactions were treated using the particle mesh Ewald method, and a cutoff of 12 Å was applied to both particle mesh Ewald9 and van der Waals interactions[53]. A time step of 2 fs was used along with the SHAKE algorithm for hydrogen atoms, and a periodic boundary condition was used. For each system, total of three replicas of 100 ns each were carried out, accumulating a total of 300 ns of simulation time. The conformations visited by the enzyme along all this simulation time were clustered based on protein backbone RMSD, and the most populated cluster was selected as a representative structure of these enzymes. The CPPTRAJ module was used to calculate the stability (structure, energy, and temperature variations), convergence (RMSD of the structures), distance, and angle of each system in the AmberTools18 software[52].

## DFT calculations

DFT calculations were performed using the Gaussian 16 package. All DFT structures were constructed based on the catalytic mechanism[54] and combined with the reaction conditions in this study. In Fig. 4c, the reason we chose $HPO_4^{2-}$ as the nucleophilic reagent to attack substrate **1** is that in alkaline environments, the phosphate buffer pair ($HPO_4^{2-}$ and $H_2PO_4^-$) in PBS buffer is mostly in the form of $HPO_4^{2-}$.

In order to ensure the rigour of the experiment, the results were calculated simultaneously for $H_2PO_4^-$ as the nucleophilic reagent to attack substrate **1**, as illustrated in Supplementary Fig. 11. Geometry optimizations of the minima and transition states involved were performed at the B3LYP-D3 level of theory with the 6-31 + G (d) basis set. Vibrational frequency calculations were performed at the same level to ensure that all stationary points were transition states (one imaginary frequency) or minima (no imaginary frequency) and to evaluate zero-point vibrational energies and thermal corrections at 298 K. Single-point energy calculations were performed at the B3LYP-D3 level using the 6-311 + G (2d, p) basis set. Solvation by water was considered using the CPCM model[55] for all of the above calculations. All Supporting computational data can be found in the Supplementary Data 2. Optimized DFT structures are illustrated with CYLView (https://www.cylview.org/).

## Reporting summary

Further information on research design is available in the Nature Portfolio Reporting Summary linked to this article.

## Data availability

Data supporting the findings of this work are available within the manuscript and Supplementary files. The datasets generated and analyzed during the current study are also available from the corresponding author. Crystal structure of *Geobacillus kaustophilus apo-Gk*OYE.8 was uploaded into the PDB database with PDB number 8X0J. Source data are provided with this paper.

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

## Acknowledgements

This work was supported by the National Key R & D Program of China 2021YFC2100100 (to J. W.), the General Program of National Natural Science Foundation of China 22378165 (to W. S.).

## Author contributions

W. S., L. W. and L. L. conceived the study and designed the experiments. L. W. performed this project. Y. W. and W. W. performed computational studies on protein dynamics. D.Y. performed the X-ray diffraction. J. H. analyzed the protein mass spectrum. J. Wen., W. W., J. Wu., Z. Z., X. C. and C. G. provided technical assistance. L. W. and W. S. wrote this manuscript. Y. Z., J. L., G. H. and X. L. revised the manuscript. All authors read, edited, and approved the final manuscript.

## Competing interests

The authors declare no competing interests.
