## [Peer Review File · Nature Communications]

Unlocking the function promiscuity of old yellow enzyme to catalyze asymmetric Morita-Baylis-Hillman reactionREVIEWER COMMENTS

Reviewer #1 (Remarks to the Author):

The authors report the results of a study on promiscuous Morita-Baylis-Hillman activities observed in several Old Yellow Enzymes. They focused on GkOYE and thoroughly modified the enzyme to significantly increase its activity, although it remained relatively low in absolute terms. The stereospecificity of the modified OYE is also only moderate.

Nevertheless, this interesting study demonstrates the breadth of enzyme catalysis and the possibility of extending the reaction space to non-natural reactions.

To corroborate their results, the authors also determined the crystal structure of one variant, GkOYE.8, and performed docking calculations and MD simulations. DFT calculations were used to rationalize the potential importance of a cysteine residue in the active site. I was asked to focus my review on the structural and computational part of the study.

According to the PDB validation report, the structure of apo-GkOYE.8 is of good quality, especially given the resolution of the diffraction dataset. However, the corresponding Table 2 in the Supplementary Information is a mess. This table is supposedly about PDB entry 8HQ2, which contains a crystal structure of human ADAM22 that has nothing to do with the enzyme studied by the authors. Furthermore, the values in the table only partially match the values in the validation report. This is sloppy at best.

A reaction intermediate (IntD) was docked to clustered snapshots from an MD trajectory. This is indeed a very reasonable strategy. However, the nature of this intermediate is quite unclear to me. In the chemical drawing in Fig. 3d, the five-membered ring appears saturated. In the neighboring PyMOL figure, the same ring is drawn with aromatic bonds. Both cannot be right, and to my understanding, both need to be corrected.

Similarly, IntD is negatively charged on the oxygen in the chemical drawing but protonated in the PyMOL figure. Which form was used for the docking calculations? The charge can make a big difference in docking. Does IntD correspond to one of the intermediates from the DFT calculations?

Did the authors consider that substrate 2 (p-nitrobenzaldehyde) could also bind similarly to typical OYE substrates via the aldehyde or even the nitro group hydrogen-bonded to H164 and H167?

What is the proposed mechanism mentioned in the captions of Fig. 4 c and d that underlies the DFT calculations? I did not quite understand where the extra (di)hydrogen phosphate comes from. Have I missed a literature reference? There are no references in this part of the manuscript (page 13, lines 265-271). Is (di)hydrogen phosphate also involved in enzyme catalysis by GkOYE? Which residue could play this role?

The authors should also tone down the claim that the DFT results “prove that residue C26 plays a catalytic role”. I see this merely as a hint. Overall, the experimental data is not entirely convincing to me either. A 70% reduction in activity observed for the Cys to Ala variant does not support a crucial role.

Reviewer #2 (Remarks to the Author):

In this work Wang et al. describe the discovery of MBH activity in old yellow enzyme (OYE) which typically catalyses a C=C bond reduction. To suppress native C=C reduction reactivity, two strategies were used:

- 1) Shutting down hydride transfer by selectively introducing mutations at positions where FMN cofactor-binding residues are found. New variants are shown to be unable to bind FMN and thus exhibit little to no reduction reactivity, increasing MBH reactivity at the same time.
- 2) Shutting down proton transfer by turning a key Y residue to an F.
- 3) Both strategies are complementary and result in higher MBH activity when combined than when used separately.

The enzyme GkOYE.8 was then subjected to rounds of directed evolution which significantly improved activity and stereoselectivity. They were even able to identify a set of only 2 mutations that directly led to reversal of stereoselectivity, with high levels of the opposite enantiomer obtained while maintaining good activity. The manuscript is largely well presented and well structured, and the refencing is thorough. However, I have a number of

comments that I believe need addressing.

1) For the substrate scope, only 5 different substrates were tested. It would be nice to see a wider range of substrates. How is activity and selectivity effected with meta-substituents on the benzaldehyde or even an undecorated benzaldehyde. A range of non-cyclic alkenes should also be tested as alkene donors such as acrylates are more commonly seen in the chemical literature. For the substrates tested, it is also seen that there is quite a drop in the selectivity. Can the authors comments on why adding a different substituent onto the aldehyde group would drop the enantioselectivity of the enzyme. It is nice however to see that their GkOYE.13 variant consistently gives the opposite enantiomer compared to their best variant.

2) With their starting enzyme GkOYE, they state in table 1 that the titre of MBH product 3 is 45.8 μM below this table they then state that in this reaction there is also 3.3-folds of reduction product 4. This is meant to be supported by supplementary figure 3. However, the numbers in figure 3 do not match the numbers in the text. For example, 3.3-folds would equate to 151 μM of 4. The SI figure however shows over 250 μM of 4. Can the authors comment on this discrepancy.

3) Additionally, in figure 2e and g the conversions of GkOYE.7 and GkOYE.8 are around 8 and 12%, respectively. However, in figure 5d they are shown to now be around 20 and 30% respectively. Can the authors comment on the different. If this is due to the fact that the reactions are ran under different pH conditions, then the conditions need to be stated in the figure legends.

4) In my view, Figure 2a is confusing and would benefit from a simpler diagram or being split into 2 figures. Firstly, when showing the mechanism for the MBH reaction, structure C doesn't need the curly arrows as C to D is a protonation step. But arrows are required on D for the deprotonation to form E. Secondly, it would be beneficial to state the positions of the histidines shown and to draw the tyrosine instead of writing Y-O-H.

5) For the molecular docking, the authors have drawn the S-enantiomer however it looks

like they have docked the R-enantiomer. The intD they have docked also is shown to be protonated and the cyclopentane ring and looks to be aromatic in the pymol presentation. Please can the authors clarify the discrepancies here.

6) Finally, the authors perform DFT calculations along the reaction coordinate for GkOYE.8, however there is no comment on what residues could potentially be stabilising the oxyanion intermediates formed during the reaction. Can the authors comment on any plausible amino acids that are stabilising these from the DFT calculations.

In summary, this paper is a good proof of concept paper. I like the story and the reasoning that led them to enhanced MBH reactivity in OYE. However, there are inconsistencies in some of their data, a lack of mechanistic rationale and a limited substrate scope. Due to this, I believe these points mean that the broader impact/significance of this work is thus limited with respect to this journal.

Reviewer #3 (Remarks to the Author):

Prof. Song and colleagues investigated the promiscuous nature of old yellow enzymes (OYEs) in facilitating the MBH reaction while inhibiting the enzyme's native reduction function through the obstruction of H⁻ and H⁺ transfer pathways. This led to the development of a mutant enzyme called GkOYE.8, which exhibited a remarkable 141.4% increase in MBH adduct yield compared to the wild-type enzyme. By conducting mutation studies and utilizing kinetic simulations, the authors gained further insights into the structural basis of GkOYE.8's ability to catalyze the MBH reaction and identify crucial residues in the active center. Following subsequent protein engineering efforts aimed at enhancing the activity and selectivity of the MBH reaction, a set of enantio-complementary mutants, namely GkOYE.11 and GkOYE.13, were successfully generated. These mutants demonstrated catalytic activity against five p-substituent aldehydes and six-membered unsaturated alkenes.

Despite the catalytic performance, this study highlights three innovative aspects: firstly, it represents the discovery of a new reactivity towards the OYE protein, capable of catalyzing the production of both (R)-MBH and (S)-MBH adducts. Secondly, the manipulation of

functional selectivity resulting in a switch of reactivity that may apply to other enzymes. Thirdly, the identification of key structural foundations to enhance catalytic efficiency and stereoselectivity, leading to the discovery of stereoselective complementary mutants GkOYE.11 and GkOYE.13 through directed evolution. Overall, I believe this is exciting work and should be appropriate for Nat. Commun. and would be of broad interest to Nat. Commun. readership.

Specific Comments:

1. While the effects of PBS buffers of different pH on the MBH reaction were explored in line 286, other buffers of corresponding pH were not investigated. It is suggested to expand on this aspect.

2. In the section: Discussing the structural bases of GkOYE.8-catalyzed MBH reaction, the potential nucleophilic catalytic residue C26 is identified. Further analysis is recommended to ascertain if such cysteine residues are also present in other old yellow enzyme sources listed in Table 1.

3. The GkOYE.8-substrate complex structure was not obtained through X-ray crystallographic analysis. Have other mutant-substrate complex crystallization experiments been conducted? Variations in protein structures might pose challenges in crystallization, thereby influencing experimental outcomes.

Some suggestions for improving the quality of the MS/Sl:

1. line 41, please rephrase "Biocatalytic C-C bond formation is an important transformation process in organic synthesis" is a little strange.

2. In line 109, the "2-cyclohexen-1-one (1)" should be "2-cyclopenten-1-one (1).

3. In fig.1(b), why not just synthesize the standard compound 3 and compare it by GC-MS.

4. In fig.2(d), the legend interpretation is not clear enough, and the column 5/6 is not complete as a blank control group.

5. In line 366, the "p-Ome" should be "p-OMe".

6. In line 385, the mutants' name should be consistent throughout the manuscript.
7. In line 395, the expression "requiring no cofactors or assistance from light" seems to have some grammar problem.
8. In line 406-407, the expression "Y169 to Phe / Trp" should be consistent throughout the manuscript.
9. The final paragraph of the discussion has too much overlap with the introduction part.
10. The circular dichroism scans mentioned in line 239 serve as supplementary verification of mutation results. To enhance clarity, it is advised to relocate this content to the Supplementary Information section.
11. It is advised to maintain consistency in the terminology used, either opting for "enantioselectivity" or "stereoselectivity", throughout the manuscript (line 329-311).

Supporting Information:

1. Why are the NMR spectra of 3c, 3d, 3e unpurified (Supplementary Fig.10, 11, 12)?
2. The HPLC chromatograms in Supplementary Fig.13, 15, 17, 19, 21 and 23 don't provide each compound's peak area data, and the determination methods for the yield are not clearly. It would be better to give a specific example of concrete compound.
3. There is no DNA sequence information of the original wild type protein GkOYE in the SI.

Response to reviewers

Response to reviewers' comments:

For your convenience, the itemized answers to the reviewers' comments are presented below (our responses are in blue font color). All of the revisions to the manuscript are highlighted in yellow.

Reviewer #1 (Remarks to the Author):

The authors report the results of a study on promiscuous Morita-Baylis-Hillman activities observed in several Old Yellow Enzymes. They focused on *GkOYE* and thoroughly modified the enzyme to significantly increase its activity, although it remained relatively low in absolute terms. The stereospecificity of the modified OYE is also only moderate. Nevertheless, this interesting study demonstrates the breadth of enzyme catalysis and the possibility of extending the reaction space to non-natural reactions. To corroborate their results, the authors also determined the crystal structure of one variant, *GkOYE.8*, and performed docking calculations and MD simulations. DFT calculations were used to rationalize the potential importance of a cysteine residue in the active site. I was asked to focus my review on the structural and computational part of the study.

Major Issues

Q1: According to the PDB validation report, the structure of apo-*GkOYE.8* is of good quality, especially given the resolution of the diffraction dataset. However, the corresponding Table 2 in the Supplementary Information is a mess. This table is supposedly about PDB entry 8HQ2, which contains a crystal structure of human ADAM22 that has nothing to do with the enzyme studied by the authors. Furthermore, the values in the table only partially match the values in the validation report. This is sloppy at best.

A1: We are sorry for the mistake. We have corrected PDB entry from "8HQ2" to "8X0J" in Table 2 of Supplementary Information (**Page 52**). After re-analyzing

the crystallization data and updating Table 2 accordingly, we confirmed it now aligns with the validation report.

Supplementary Table 2. Data collection and refinement statistics of *apo-GkOYE.8* crystal structure.

	Apo-GkOYE.8
PDB code	8X0J
Data collection	
Space group	I 21 21 21
Cell dimension	
a, b, c (Å)	86.06, 143.57, 155.47
α , β , γ (Å)	90.00, 90.00, 90.00
Resolution (Å)	45.74 - 3.11 Å
R _{merge}	0.17 (0.49)
R _{pim}	0.146 (0.406)
CC1/2	0.967(0.726)
I/ σ (I)	5.8 (2.1)
% Data completeness (in resolution range)	98.5(45.74-3.11)
Refinement	
No. reflections	17753
Rwork/Rfree	0.2151/0.2648
Total number of atoms	5314
Wilson B-factor (Å ²)	39.8
R.m.s.deviations	
bond lengths (Å)	0.0063
bond angles (°)	1.4525
Ramachandran outliers (%)	0.00
Ramachandran favored (%)	96.15

Q2: A reaction intermediate (IntD) was docked to clustered snapshots from an MD trajectory. This is indeed a very reasonable strategy. (1) However, the nature of

this intermediate is quite unclear to me. (2) In the chemical drawing in Fig. 3d, the five-membered ring appears saturated. In the neighboring PyMOL figure, the same ring is drawn with aromatic bonds. Both cannot be right, and to my understanding, both need to be corrected.

A2: Thanks for your kind suggestions. We will respond to your comments in the following two aspects:

- (1) Regarding the confusion about the intermediate structure (IntD), we apologize for the unclear description. During molecular docking, the structure of product **3**, not the IntD, was docked into the catalytic pocket of *GkOYE.8* to identify residues within a 4 Å radius. IntD is a conceptual structure, representing the covalent intermediate formed in the MBH reaction. Upon analysis, its structure, apart from the catalyst part, is most similar to that of product **3**. Thus, product **3** was chosen for docking. This explanation is also provided in lines 223-224 of the revised manuscript. Consequently, the small molecule depicted in the PyMOL diagram on the right represents the structure of product **3**.
- (2) Regarding the discrepancy in the five-membered ring structure in the PyMOL diagram, it's due to our erroneous operation of the software. We have adjusted the display mode in the PyMOL diagram in Fig. 3d to present it more clearly.

Fig. 3d The architecture of the complex form of *GkOYE.8-3* and the key residues within a 4 Å radius.

Q3: (1) Similarly, IntD is negatively charged on the oxygen in the chemical drawing but protonated in the PyMOL figure. Which form was used for the docking calculations? The charge can make a big difference in docking. (2) Does IntD correspond to one of the intermediates from the DFT calculations?

A3: Thanks for your kind suggestions.

- (1) The confusion arises from an error in our description, which has been clarified in question 2. Our docking structure is actually product **3**, not IntD. Consequently, the small molecule depicted in the PyMOL diagram represents the structure of product **3**. The charge of the molecule during docking can impact the results, but our docking limit was based on specific distances within product **3**. Consequently, the residues within a radius of 4 Å around product **3** remained unaltered across the various docking outcomes.
- (2) IntD represents a conceptual structure, not a specific intermediate from DFT calculations. In Figure 4c, IntD corresponds to the structure of Int 2, and in Figure 4d IntD corresponds to the structure of Int 2'. We have annotated this clarification in Figures 4c and 4d as per your reminder.

Q4: Did the authors consider that substrate **2** (*p*-nitrobenzaldehyde) could also bind similarly to typical OYE substrates via the aldehyde or even the nitro group hydrogen-bonded to H164 and H167?

A4: Thanks for your constructive suggestion. We did consider the possibility that the aldehyde and nitro groups in *p*-nitrobenzaldehyde could bind to H164 and H167. To address this, we used an excess of substrate **1** relative to substrate **2** in the reaction. The K_m values of the two substrates were measured, showing that substrate **1** has a smaller K_m and therefore a stronger binding affinity to the protein compared to substrate **2** (*p*-nitrobenzaldehyde). Your suggestion will guide our future studies, and we plan to optimize the reaction process by adding substrate **1** sequentially and preferentially to ensure it binds to the protein first.

Q5: (1) What is the proposed mechanism mentioned in the captions of Fig. 4 c and d that underlies the DFT calculations? (2) I did not quite understand where the extra (di)hydrogen phosphate comes from. Have I missed a literature reference? There are no references in this part of the manuscript (page 13, lines 265-271). (3) Is (di)hydrogen phosphate also involved in enzyme catalysis by *GkOYE*? Which residue could play this role?

A5: Thanks for your kind suggestions.

(1) Based on previous literature reports^{35,36}, we proposed the putative mechanism of the MBH reaction in this study. As shown in the Figures 4c and 4d, the putative mechanism of MBH reaction consists of 5 steps. The step 1: Michael addition. The step 2: Aldol reaction. The step 3 and 4: Proton transfer. The step 5: Elimination.

Fig. 4 The study on the nucleophilic catalytic residue of C26. (c) The proposed mechanism and Gibbs free energy profile of MBH reaction without enzyme calculated by DFT. The putative mechanism of MBH reaction consists of 5 steps^{35,36}. The step 1: Michael addition. HPO_4^{2-} as a nucleophile attacks substrate **1**³⁷. The step 2: Aldol reaction. The step 3 and 4: Proton transfer. The step 5: Elimination. (d) The proposed mechanism and Gibbs free energy profile of MBH reaction with a theozyme model calculated by DFT. The theozyme model includes substrate **1**, substrate **2**, water, methanethiol (substituting for residue C26) and acetic acid (substituting for residue E59). DFT-computed Gibbs free energies (in kcal/mol) at theCPCM(water)/B3LYP-D3/6-311++G(2d,p)//CPCM(water)/B3LYP-D3/6-31+G(d) level of theory and transition-state structures (carbon: gray, hydrogen: white, oxygen: red, nitrogen: blue, sulfur: yellow, phosphorus: orange and distances are shown in Å).

(2) According to literature reports, (di)hydrogen phosphate can be involved in

reaction as a nucleophile³⁷. In addition, (di)hydrogen phosphate and water molecules present in PBS buffer can be used to regulate the proton transfer of MBH reaction. In addition, we found that the MBH reaction in PBS buffer was more active than that in pure water in this study. Therefore, we calculated and compared the energies of these two proton transfer modes separately in our calculations. As shown in Supplementary Figure 10, the results indicated that mode of phosphate anion is more exothermic than water, thus confirming the phosphate anion may accelerate the proton transfer. In the model of the background reaction without enzyme, it is taken into account that in an alkaline environment, the phosphate buffer pairs (HPO_4^{2-} and H_2PO_4^-) in PBS buffer are mostly present in the form of HPO_4^{2-} . Therefore, we mainly calculated the energy barrier of attacking substrate **1** with HPO_4^{2-} as a nucleophilic reagent, see Fig. 4. And the results of attacking substrate **1** with H_2PO_4^- as a nucleophilic reagent are shown in Supplementary Fig. 11.

- (3) The (di)hydrogen phosphate has two functions, the first one is that it can participate in MBH reaction as a nucleophilic reagent in step 1 of the chemocatalytic mechanism of Figure 4c and has been reported³⁷. And in the enzyme catalysis mechanism, C26 acts as a nucleophilic reagent to participate in the reaction. The second one is that it can participate in the proton transfer reaction in Figure 4c and Figure 4d (step 3 and step 4). The (di)hydrogen phosphate is involved in the enzyme catalysis of *GkOYE*. Since (di)hydrogen phosphate is a component of the external environment involved in the enzyme-catalyzed proton transfer process and this process was reported by Anthony P. Green to be mediated by water molecules²³. So we hypothesise that the process of proton transfer in the MBH reaction is assisted by components of the environment. Based on the available results, we have carefully considered the possibility that there are no residues in *GkOYE* that could play the role of (di)hydrogen phosphate.

Supplementary Figure 10. DFT calculations were conducted to compare the energies of the two proton transfer modes. The black line represents the proton transfer involving phosphate anion, while the rose line indicates a water-mediated proton transfer. The results indicated that mode of phosphate anion is more exothermic than water, thus confirming the phosphate anion may accelerate the proton transfer.

Q6: The authors should also tone down the claim that the DFT results “prove that residue C26 plays a catalytic role”. I see this merely as a hint. Overall, the experimental data is not entirely convincing to me either. A 70% reduction in activity observed for the Cys to Ala variant does not support a crucial role.

A6: Thanks for your constructive suggestions.

- (1) We agree that DFT calculations alone cannot conclusively determine the role of C26. We have modified the original statement as: "Furthermore, quantum mechanics (QM) calculations employing density functional theory (DFT) also suggest that residues C26 and E59 may be key residues in the reaction process." Please refer to the revised manuscript (**page 13, lines 261-262**).
- (2) Regarding wet experimental results, we observed a significant reduction in activity (more than 60%) for 19 mutant proteins with saturation mutagenesis of C26. We also noted that only cysteine at position 26 yielded high activity. We then analyzed why the reduction didn't exceed 90%. First, there is a background reaction in the control (Figure 3g). Second, enzymes proteins may accelerate the reaction by locally concentrating the substrates. Additionally, mass spectrometry

data (Figures 4a and 4b) showed no intermediate B formation after the mutation of C26 to Ala, suggesting C26 acts as a nucleophilic reagent in the MBH reaction.

Reviewer #2 (Remarks to the Author):

In this work Wang et al. describe the discovery of MBH activity in old yellow enzyme (OYE) which typically catalyses a C=C bond reduction. To suppress native C=C reduction reactivity, two strategies were used:

- 1) Shutting down hydride transfer by selectively introducing mutations at positions where FMN cofactor-binding residues are found. New variants are shown to be unable to bind FMN and thus exhibit little to no reduction reactivity, increasing MBH reactivity at the same time.
- 2) Shutting down proton transfer by turning a key Y residue to an F.
- 3) Both strategies are complementary and result in higher MBH activity when combined than when used separately.

The enzyme *GkOYE.8* was then subjected to rounds of directed evolution which significantly improved activity and stereoselectivity. They were even able to identify a set of only 2 mutations that directly led to reversal of stereoselectivity, with high levels of the opposite enantiomer obtained while maintaining good activity. The manuscript is largely well presented and well structured, and the referencing is thorough. However, I have a number of comments that I believe need addressing.

Major Comments

Q1: (1) For the substrate scope, only 5 different substrates were tested. It would be nice to see a wider range of substrates. How is activity and selectivity effected with meta-substituents on the benzaldehyde or even an undecorated benzaldehyde. A range of non-cyclic alkenes should also be tested as alkene donors such as acrylates are more commonly seen in the chemical literature. (2) For the substrates tested, it is also seen that there is quite a drop in the selectivity. Can the authors comments on why adding a different substituent onto the aldehyde group would drop the enantioselectivity of the enzyme. It is nice however to see that their *GkOYE.13* variant consistently gives the opposite enantiomer compared to their best variant.

A1: Thanks for your constructive suggestions.

- (1) We expanded the substrate scope to include undecorated aromatic aldehyde **2e**, *meta*-substituted aromatic aldehyde substrates **2f** and **2j**, and double-substituted aromatic aldehyde substrates **2h** and **2i**. Additionally, non-cyclic alkenes such as methyl acrylate, but-3-en-2-one, pent-1-en-3-one, and aliphatic aldehydes propionaldehyde, butyraldehyde, pentanal and hexanal were tested. The results outlined in Fig. 6 demonstrate that mutants *GkOYE.11* and *GkOYE.13* effectively catalyze MBH reactions with substrates **2e-2i**. Interestingly, the undecorated product (**3f**) exhibits lower yields compared to the *para*-electron-withdrawing substituent products (**3**, **3a-3c**) but higher yields compared to the *para*-electron-donating substituent product (**3d**). Surprisingly, it is notable that when *GkOYE.11* and *GkOYE.13* catalyze the undecorated aromatic aldehyde (**2e**) and the *meta*-substituted aromatic aldehydes (**2f** and **2j**), there is a surprising stereoselective flip observed in the enantiomeric ratio of the product. This flip occurs despite the expected outcome based on the type of mutant used, indicating a complex and nuanced stereoselectivity pattern in these reactions. It can be seen that the stereoselectivity of the products varied based on the position and type of substituent. The obtained mutants *GkOYE.11* and *GkOYE.13* are more active toward aromatic substrates and inactive toward the tested non-cyclic alkenes, likely due to their increased affinity for aromatic substrates during directed evolution. Detailed NMR, HPLC, and GC-MS results are now included in Figures 18-22, 35-44 and 50-54 of the Supplementary Information.
- (2) The size and configuration of the *para*-substituents significantly impact the enantiomeric ratio in the reaction. Larger substituents tend to yield higher enantiomeric ratios, as observed with substrates **2**, **2a** and **2b** gradually. This indicates that large substituents are more conducive to the generation of chiral products catalyzed by *GkOYE.11* and *GkOYE.13*.

Fig. 6 Substrate scope of *GkOYE.11* and *GkOYE.13*. Biotransformations were performed using aldehyde (1 mM), activated alkene (5 mM) and catalyst (100 μ M). The detailed information of results and determination methods for the yield and enantiomeric ratio of the MBH adducts were shown in Supplementary materials.

Q2: With their starting enzyme *GkOYE*, they state in table 1 that the titer of MBH product 3 is 45.8 μ M below this table they then state that in this reaction there is also 3.3-folds of reduction product 4. This is meant to be supported by

supplementary figure 3. However, the numbers in figure 3 do not match the numbers in the text. For example, 3.3-folds would equate to 151 μM of 4. The SI figure however shows over 250 μM of 4. Can the authors comment on this discrepancy.

A2: We apologize for the mistake. The titer of MBH product **3** is indeed 45.8 μM , and the titer of product **4** is 280.4 μM as shown in Supplementary Figure 3. Therefore, the correct multiple should be 6.3, not 3.3. We have corrected this information in the revised manuscript (**lines 151**) and the updated Supplementary Figure 3 accordingly.

Q3: Additionally, in figure 2e and g the conversions of *GkOYE.7* and *GkOYE.8* are around 8 and 12%, respectively. However, in figure 5d they are shown to now be around 20 and 30% respectively. Can the authors comment on the different. If this is due to the fact that the reactions are ran under different pH conditions, then the conditions need to be stated in the figure legends.

A3: Thanks for your kind suggestions. The differences in yields for *GkOYE.7* and *GkOYE.8* in Figures 2e and 2g compared to Figure 5d are indeed due to variations in reaction conditions, particularly the buffer pH. The reactions in Figures 2e and 2g were conducted during the pre-blocking of the natural reaction, where pH was 7.4 and not optimized. However, in Fig. 5d, optimal reaction conditions (pH 10) were used, resulting in higher yields. We have updated the legend for Fig. 5d to include the specific reaction conditions for clarity.

Q4: In my view, Figure 2a is confusing and would benefit from a simpler diagram or being split into 2 figures. Firstly, when showing the mechanism for the MBH reaction, structure C doesn't need the curly arrows as C to D is a protonation step. But arrows are required on D for the deprotonation to form E. Secondly, it would be beneficial to state the positions of the histidines shown and to draw the tyrosine instead of writing Y-O-H.

A4: Thanks for your kind suggestions. We have simplified Figure 2a as per your

recommendation. Additionally, we have included the specific MBH reaction mechanism in Figures 4c and 4d of the revised manuscript. The arrow notation in the mechanism writing section has been standardized as suggested. Additionally, we replaced "Y-O-H" with "tyrosine" in Figure 2a and labeled the precise positions of histidine as you suggested.

Fig. 2 Enhancing MBH function of *GkOYE* from the perspective of catalytic mechanism.

(a) The mechanism of natural reduction reaction catalyzed by OYEs and the putative mechanism of MBH reaction in this study. The mechanism of MBH reaction was inferred based on the reported mechanisms of enzyme catalysis²³ and chemical small molecule catalysis²⁸. Nu: The nucleophilic catalytic residue in *GkOYE* protein. IntD: the structure of covalently connected intermediate of nucleophilic catalytic residue with **1** and **2**.

Q5: For the molecular docking, the authors have drawn the *S*-enantiomer however it looks like they have docked the *R*-enantiomer. The IntD they have docked also is shown to be protonated and the cyclopentane ring and looks to be aromatic in the pymol presentation. Please can the authors clarify the discrepancies here.

A5: Thanks for your constructive suggestions.

(1) We apologize for our unclear description. During molecular docking, we docked the structure of product **3**, not the IntD, into the catalytic pocket of *GkOYE*.⁸ This was done to identify residues within a 4 Å radius. The PyMOL diagram in

Figure 3d represents the structure of product **3** with an *R*-type configuration and a hydroxyl group attached to the chiral carbon in its protonated form. We have modified and annotated Figure 3d accordingly.

Fig. 3d The architecture of the complex form of *GkOYE.8-3* and the key residues within a 4 Å radius.

(2) Regarding the discrepancy in the PyMOL diagram, it's due to our erroneous operation of the software. We acknowledge that the initial display mode was susceptible to misinterpretation, so we have adjusted the display mode in the revised manuscript to present a clearer mode representation in Figure 3d.

Fig. 3d The architecture of the complex form of *GkOYE.8-3* and the key residues within a 4 Å radius.

Q6: Finally, the authors perform DFT calculations along the reaction coordinate for *GkOYE.8*, however there is no comment on what residues could potentially be stabilising the oxanion intermediates formed during the reaction. Can the authors comment on any plausible amino acids that are stabilising these from the DFT calculations.

A6: Thanks for your constructive suggestions. Our DFT calculations were performed

using a theozyme model that included substrate **1**, substrate **2**, water, methanethiol (substituting for residue C26) and acetic acid (substituting for residue E59). However, other residues in the vicinity of the substrate molecule were not included in these calculations, as explained on **page 16, lines 306-308**. Regarding potential amino acids that could stabilize the oxyanion intermediates, it's important to note that the real substrate-enzyme complex structure was unavailable, and MD simulations did not identify any residues that could form a hydrogen-bonding interactions with the oxyanion. Therefore, we deduced that the residues stabilizing the oxyanion might not be present in this reaction model. This absence could contribute to the lower catalytic efficiency observed in the reaction.

In summary, this paper is a good proof of concept paper. I like the story and the reasoning that led them to enhanced MBH reactivity in OYE. However, there are inconsistencies in some of their data, a lack of mechanistic rational and a limited substrate scope. Due to this, I believe these points mean that the broader impact/significance of this work is thus limited with respect to this journal.

A: Thank you for your interest in our work and for providing valuable suggestions.

We have carefully revised and corrected the data inconsistencies, reorganized the reaction mechanism and DFT calculations, and expanded the substrate scope in our experiments. We believe these improvements address the limitations and enhance the significance of our work, aligning it more closely with the standards of the journal.

Reviewer #3 (Remarks to the Author):

Prof. Song and colleagues investigated the promiscuous nature of old yellow enzymes (OYEs) in facilitating the MBH reaction while inhibiting the enzyme's native reduction function through the obstruction of H^- and H^+ transfer pathways. This led to the development of a mutant enzyme called *GkOYE.8*, which exhibited a remarkable 141.4% increase in MBH adduct yield compared to the wild-type enzyme. By conducting mutation studies and utilizing kinetic simulations, the authors gained further insights into the structural basis of *GkOYE.8*'s ability to catalyze the MBH reaction and identify crucial residues in the active center. Following subsequent protein engineering efforts aimed at enhancing the activity and selectivity of the MBH reaction, a set of enantio-complementary mutants, namely *GkOYE.11* and *GkOYE.13*, were successfully generated. These mutants demonstrated catalytic activity against five *p*-substituent aldehydes and six-membered unsaturated alkenes.

Despite the catalytic performance, this study highlights three innovative aspects: firstly, it represents the discovery of a new reactivity towards the OYE protein, capable of catalyzing the production of both (*R*)-MBH and (*S*)-MBH adducts. Secondly, the manipulation of functional selectivity resulting in a switch of reactivity that may apply to other enzymes. Thirdly, the identification of key structural foundations to enhance catalytic efficiency and stereoselectivity, leading to the discovery of stereoselective complementary mutants *GkOYE.11* and *GkOYE.13* through directed evolution. Overall, I believe this is exciting work and should be appropriate for Nat. Commun. and would be of broad interest to Nat. Commun. readership.

Specific Comments:

Q1: While the effects of PBS buffers of different pH on the MBH reaction were explored in line 286, other buffers of corresponding pH were not investigated. It is suggested to expand on this aspect.

A1: Thanks for your constructive suggestions. We investigated the effects of carbonate buffer and glycine-NaOH buffer on the MBH reaction catalyzed by

GkOYE.8. The results showed that while both buffers yielded higher MBH product **3** compared to PBS buffer, they also led to significantly higher background reaction rates (yielding 51.4% and 46.0%, respectively). This high background reaction resulted in a substantial reduction in the enantiomeric ratio of (*R*)-**3** to 50:50. Therefore, we concluded that PBS buffer at pH 10.0 is more suitable for *GkOYE.8* catalyzed MBH reactions. We have included these findings in the revised manuscript (lines 290-295) and the Supplementary Figure 12.

Supplementary Figure 12. The effect of different buffers at pH=10 on the MBH reaction was investigated. The reactions were carried out using 5 mM **1**, 1 mM **2**, and 100 μ M purified protein in different buffer (pH 10.0) with 3% methanol (MeOH) as a co-solvent. Control: Background reaction without enzyme.

Q2: In the section: Discussing the structural bases of *GkOYE.8*-catalyzed MBH reaction, the potential nucleophilic catalytic residue C26 is identified. Further analysis is recommended to ascertain if such cysteine residues are also present in other old yellow enzyme sources listed in Table 1.

A2: Thank you for your suggestions. Upon comparing the amino acid sequences of the five proteins listed in Table 1, we found that residue C26 is conserved in NemA, XenA, and *GkOYE*. In GluER and MR, residue 26 is Thr, which can also

act as a nucleophile attacking the β C of the double bond of substrate **1**. We have included this information in the revised manuscript (lines 241-244) and the Supplementary Figure 8.

Supplementary Figure 8. The sequence alignment results of different old yellow enzymes.

Q3: The *GkOYE.8*-substrate complex structure was not obtained through X-ray crystallographic analysis. Have other mutant-substrate complex crystallization experiments been conducted? Variations in protein structures might pose challenges in crystallization, thereby influencing experimental outcomes.

A3: Thank you for your suggestions. We have indeed conducted crystallization experiments on mutant-substrate complexes, such as *GkOYE.10*, *GkOYE.11*, and *GkOYE.13*. Despite efforts to optimize the conditions, these attempts were

unsuccessful. This challenge may stem from the protein's low affinity for the substrate, which can affect the crystallization outcome.

Some suggestions for improving the quality of the MS/SI:

Q1: line 41, please rephrase “Biocatalytic C-C bond formation is an important transformation process in organic synthesis” is a little strange.

A1: Thank you for pointing that out. We've revised the phrase as per your suggestion. We have corrected “Biocatalytic C-C bond formation is an important transformation process in organic synthesis” to “C-C bond formation is a crucial process in organic synthesis” in the revised manuscript (now in **line 33**).

Q2: In line 109, the “2-cyclohexen-1-one (**1**)” should be “2-cyclopenten-1-one (**1**).

A2: We are sorry for the mistake. We have corrected “2-cyclohexen-1-one (**1**)” to “2-cyclopenten-1-one (**1**) in the revised manuscript (now in **line 101**).

Q3: In fig.1(b), why not just synthesize the standard compound **3** and compare it by GC-MS.

A3: Thank you for your kind suggestion. The GC-MS method for comparing the newly generated substance to product **3** is simpler and more direct. However, we wanted to prove the structure of the new substance as product **3** using NMR data. Therefore, we carried out the isolation and purification.

Q4: In fig.2(d), the legend interpretation is not clear enough, and the column 5/6 is not complete as a blank control group.

A4: Thanks for your suggestion. Here's the revised legend for Figure 2d based on your suggestions:

"Comparison of reduction activity between *GkOYE* and variant *GkOYE.7*. Columns 1 and 2 represent the reduction activity of *GkOYE.7* with and without FMN. Columns 3 and 4 represent the reduction activity of *GkOYE* with and without

FMN. Columns 5-8 represent blank control groups. "+" indicates presence and "-" indicates absence. The added components are 5 mM **1**, 1 mM NADPH, 1 mM FMN, and 100 μ M purified protein in PBS (pH 7.4) with 1% methanol (MeOH) as a co-solvent."

Additionally, blank control experiments were extended to include experimental groups 6 and 7 to investigate the impact of NADPH and FMN on the reduction reaction of substrate **1**, respectively. The results demonstrate that NADPH and FMN do not act as catalysts for the reduction of substrate **1**.

Q5: In line 366, the "*p*-Ome" should be "*p*-OMe".

A5: Thank you for your kind suggestion. We have corrected the "*p*-Ome" to "*p*-OMe". (now in **line 377**).

Q6: In line 385, the mutants' name should be consistent throughout the manuscript.

A6: We are sorry for the mistake. We have corrected the mutants' name in the revised manuscript (now in **line 403 and line 404**).

Q7: In line 395, the expression "requiring no cofactors or assistance from light" seems has some grammar problem.

A7: Thank you for your kind suggestion. We have corrected the grammar of this sentence. The correct statement is "In comparison, this reaction demonstrates an unexpected promiscuous function of ene-reductases for C-C bond formation, which does not require cofactors or assistance from light." (now in **lines 413-414**)

Q8: In line 406-407, the expression "Y169 to Phe / Trp" should be consistent throughout the manuscript.

A8: Thank you for your kind suggestion. The expression 'Y169 to Phe/Trp' was consistently used throughout the manuscript, which is in line with the statement made in **lines 340-344**.

Q9: The final paragraph of the discussion has too much overlap with the introduction part.

A9: Thank you for your kind suggestion. The final paragraph of the discussion has been revised to eliminate any redundancy with the introduction. For more information, refer to **lines 436-437** of the revised manuscript.

Q10: The circular dichroism scans mentioned in line 239 serve as supplementary verification of mutation results. To enhance clarity, it is advised to relocate this content to the Supplementary Information section.

A10: Thank you for your suggestion. We have relocated this content to the Supplementary Information section, see Supplementary Figure 7 for details.

Q11: It is advised to maintain consistency in the terminology used, either opting for "enantioselectivity" or "stereoselectivity", throughout the manuscript (line 329-311).

A11: Thank you for your kind suggestion. We have corrected "enantioselectivity" to "stereoselectivity" in the revised manuscript (**line 339-340**).

Supporting Information:

Q1: Why are the NMR spectras of 3c, 3d, 3e unpurified (Supplementary Fig.10, 11, 12)?

A1: We are sorry for the mistake. To refine the experimental data, we further isolated and purified these compounds. The purity of the products significantly improved after NMR verification. Supplementary Figures 15, 16, and 17 display the results.

Q2: The HPLC chromatograms in Supplementary Fig.13, 15, 17, 19, 21 and 23 don't provide each compound's peak area data, and the determination methods for the yield are not clearly. It would be better to give a specific example of concrete

compound.

A2: Thank you for your kind suggestion. The peak area data for Supplementary Fig. 23, 25, 27, 29, 31 and 33 has been supplemented. Additionally, the method for determining yield and the corresponding calculation formula have been added to the Methods section of the revised Manuscript (**lines 498-500**). For example, the yield of **3** can be obtained by the equation:

$$\text{yield} = C(3) * \text{dilution factor} / C(2) * 100\% = 0.417 * 2/1 * 100\% = 83.4\%$$

Q3: There is no DNA sequence information of the original wild type protein *GkOYE* in the SI.

A3: Thank you for your kind suggestion. Because the *GkOYE* gene (from *Geobacillus kaustophilus*) was synthesized by GenScript and codon optimized for *E. coli*, so we gave the optimized DNA sequence in Supplementary Note1 (**page 61**).

REVIEWERS' COMMENTS

Reviewer #1 (Remarks to the Author):

In their response letter, the authors adequately answered and clarified the issues I had raised in the original submission. I am satisfied. The quality of the manuscript has increased significantly, also due to the corrections and additions made based on the other reviewers' comments.

Reviewer #2 (Remarks to the Author):

The authors have addressed my initial comments to a satisfactory level and I am happy for this paper to be submitted into Nature Communications.

Reviewer #3 (Remarks to the Author):

The authors have made sustainable revisions, and my comments have been well addressed. At this stage, this reviewer recommends publishing the manuscript in Nature Communications.

A minor suggestion is that most of the substrates in Fig. 6 could be displayed in the Supplementary Information. As suggested by reviewer-2, more substrate examples have been showcased in the revised Fig. 6. However, the reaction performances appear somewhat unoptimized. Considering that this work is a proof-of-concept study for a new enzyme reactivity rather than a synthetic methodological work, this reviewer would support its acceptance.

Response to reviewers

Reviewer #1 (Remarks to the Author):

In their response letter, the authors adequately answered and clarified the issues I had raised in the original submission. I am satisfied. The quality of the manuscript has increased significantly, also due to the corrections and additions made based on the other reviewers' comments.

A: Thank you very much for your valuable suggestions on improving the quality of the article.

Reviewer #2 (Remarks to the Author):

The authors have addressed my initial comments to a satisfactory level and I am happy for this paper to be submitted into Nature Communications.

A: Thank you very much for your constructive suggestions, especially in the sections on extending the substrate scope and mechanism calculation, which are very helpful to improve the quality of the article.

Reviewer #3 (Remarks to the Author):

The authors have made sustainable revisions, and my comments have been well addressed. At this stage, this reviewer recommends publishing the manuscript in Nature Communications.

A minor suggestion is that most of the substrates in Fig. 6 could be displayed in the Supplementary Information. As suggested by reviewer-2, more substrate examples have been showcased in the revised Fig. 6. However, the reaction performances appear somewhat unoptimized. Considering that this work is a proof-of-concept study for a

new enzyme reactivity rather than a synthetic methodological work, this reviewer would support its acceptance.

A: Thank you very much for the valuable suggestions and the recognition of our study. We have followed your suggestion to present some of the data in Fig. 6 into the Supplementary Information.